# A single-cell atlas of *Drosophila* trachea reveals glycosylation-mediated Notch signaling in cell fate specification

Yue Li [1,2], Tianfeng Lu[1,2,5], Pengzhen Dong [1,2,5], Jian Chen[1,2], Qiang Zhao[1,2], Yuying Wang[1,2], Tianheng Xiao[1,2], Honggang Wu [3] ✉, Quanyi Zhao [4] ✉ & Hai Huang [1,2] ✉

The *Drosophila* tracheal system is a favorable model for investigating the program of tubular morphogenesis. This system is established in the embryo by post-mitotic cells, but also undergoes remodeling by adult stem cells. Here, we provide a comprehensive cell atlas of *Drosophila* trachea using the single-cell RNA-sequencing (scRNA-seq) technique. The atlas documents transcriptional profiles of tracheoblasts within the *Drosophila* airway, delineating 9 major subtypes. Further evidence gained from in silico as well as genetic investigations highlight a set of transcription factors characterized by their capacity to switch cell fate. Notably, the transcription factors Pebbled, Blistered, Knirps, Spalt and Cut are influenced by Notch signaling and determine tracheal cell identity. Moreover, Notch signaling orchestrates transcriptional activities essential for tracheoblast differentiation and responds to protein glycosylation that is induced by high sugar diet. Therefore, our study yields a single-cell transcriptomic atlas of tracheal development and regeneration, and suggests a glycosylation-responsive Notch signaling in cell fate determination.

*Drosophila* harbors a ramifying network of epithelial tracheal tubes that serves as a combined respiratory and circulatory system, supporting the essential aerobic respiration required for animal locomotion and tissue functions. The branching pattern of the tracheal network is generated during embryogenesis, which involves cell shape changes, differentiation, and migration, as well as branch fusion[1]. The epithelial sheets of tracheal pits that arise from tracheal primordia invaginate, elongate, and ramify into complex tubular structures, and these various cellular behaviors follow a spatiotemporally stereotyped program[2]. The *Drosophila* tracheal tubes are developmentally akin to mammalian blood vessels and are established in a manner similar to the branching morphogenesis of many other tubular organs such as the lung, kidney, and mammary gland[3]. Whereas the mammalian airway has been extensively characterized as composed of diverse cell types, the major populations of *Drosophila* tracheal primordia are ambiguously annotated as epithelial tracheoblasts.

Genetic studies in *Drosophila* have provided a wealth of information regarding the molecular events of tracheal morphogenesis[4,5]. The fibroblast growth factor (FGF) and the FGF receptor, encoded by *Drosophila branchless* (*bnl*) and *breathless* (*btl*) genes respectively, are pivotal determinants of the primary branching process. Animals defective for *btl* fail to generate the complete branching pattern and exhibit selective tracheal loss[6,7], phenocopying loss-of-function of *bnl*[8]. Misexpression of *bnl* in the embryo triggers primary branches outgrowth and directs branches towards positions with ectopic Bnl protein[8]. Genetic perturbations in larval tissues demonstrate

[1]Department of Cell Biology, and Second Affiliated Hospital, Zhejiang University School of Medicine, Hangzhou, Zhejiang Province 310058, China. [2]Zhejiang Provincial Key Laboratory of Genetic & Developmental Disorders, Zhejiang University School of Medicine, Hangzhou, Zhejiang Province 311121, China. [3]Women's Hospital, Zhejiang University School of Medicine, Hangzhou, Zhejiang Province 310058, China. [4]Division of Cardiovascular Medicine and Cardiovascular Institute, School of Medicine, Stanford University, 300 Pasteur Drive, Falk CVRC, Stanford, CA 94305, USA. [5]These authors contributed equally: Tianfeng Lu, Pengzhen Dong. ✉e-mail: honggangwu@zju.edu.cn; quanyiz@stanford.edu; haihuang@zju.edu.cn

consistent roles of FGF/FGFR in tracheal growth and branching[7,9], indicating a continuous fundamental role of FGF signaling in tracheal morphogenesis. Tracheal identity also depends on the function of Trachealess (Trh), a bHLH-PAS transcription factor with early tracheal-specific expression. Deficiency of Trh abolishes the development of the tracheal system[5,10,11]. Among the regulators that sculpt the *Drosophila* tracheal network, cross-regulatory interactions between Decapentaplegic (Dpp), Wingless (Wg), and epidermal growth factor (EGF) signaling also contribute to the commitment of tracheal branches in both embryo and larva[12–16]. One of the early morphogenetic events of *the Drosophila* trachea system is the specification and growth of primary branches, which follow a stereotyped pattern (Fig. 1a) through the action of specific genes. For instance, Dpp produced in ectodermal cells promotes the development of the dorsal branch (DB), while EGF is required for the formation of the dorsal trunk (DT)[16,17]. *knirps* (*kni*) is expressed in DB and controls tracheal cell migration and formation of DB downstream of Dpp signaling[18,19]. *spalt* (*sal*) shows prominent expression in DT and is responsible for the specification of dorsal cell population[20]. The mutual repression between Kni and Sal is necessary for terminal branch formation. Spiracular branch (SB) represents a heterogeneous cell population and expresses a set of transcription factors including Pebbled (Peb), Cut (Ct), and Escargot (Esg)[21,22]. Individual branches corresponding to domains of gene expression respect lineage restriction boundaries that are influenced by the activation of Notch signaling[19], which also specifies individual tracheal cell fates within groups of equivalent cells[23].

Metamorphosis at the late larval stage triggers extensive histolysis and a reorganization of the *Drosophila* trachea, which has been employed as a unique model system to study tubular organ formation/regeneration, especially at the interface of cellular signaling and metabolic conditions. Unlike in the embryo where the tracheal formation is initiated by groups of post-mitotic cells that undertake the programs of tubulogenesis and branching morphogenesis, larval tracheoblasts extensively remodel a pre-existing framework to shape the adult airway. During larval stages, tracheoblasts are arrested in G2 and are safeguarded by the ATR/Chk1 kinases[24]. By the end of the larval stage at L3, contrasting to cells in other tracheal metameres that undergo multiple cycles of endoreplication, the cells constituting the second metamere (Tr2) escape from Fzr-mediated endoreplication by expression of String/Cdc25, reenter the mitotic program and contribute to adult trachea[25,26]. In both *Drosophila* and higher organisms, the establishment of the tracheal system is intricately modulated by systemic and environmental signals. Nutrition has been suggested to play an integral role in tubulogenesis and angiogenesis. For instance, high-fat diet promotes migration of endothelial cells and tubulogenesis[27], and tissue vascularization, in turn, fine-tunes metabolic homeostasis under high-fat consumption[28]. Conversely, high-sugar treatment attenuates endothelial cell migration and tubulogenesis[29]. In *Drosophila*, the systemic insulin-like neuropeptides shape the growth of specific tracheal subsets and potentiate tracheal branching[30]. Correspondingly, insulin signaling negatively regulates the migration of tracheal progenitors[31]. Despite these findings, the extent to which dietary conditions modulate tubulogenesis remains vague at the cellular and molecular level.

Given the diverse behaviors and the morphological and genetic indications of heterogeneity of the tracheoblasts, a comprehensive visualization of molecular signatures, preferentially at single-cell resolution, is important for a deeper understanding of *Drosophila* tracheal morphogenesis. Here, we describe the single-cell atlas of *Drosophila* airway, elucidate the transcriptomes and transcriptional regulatory networks of its major tracheal cell types, and identify key regulators responsible for cell identity determination. The atlas reveals that two clusters of tracheal cells, the dorsal branch (DB) and progenitor cells (PC), characterized by their heterogeneity and differentiation potential, initiate a developmental trajectory towards diverse tracheal cells. Further analysis shows that principal determinants of PC, DB, and DT populations as well as key drivers in this trajectory are dependent on Notch signaling. The activity of Notch signaling in the trachea exhibits temporospatial kinetics, which vanishes along developmental pseudotime and varies in distinct populations. Further evidence suggests that Notch signaling is promoted by protein glycosylation and is responsive to a high-sugar diet, which provides mechanistic insights into differentiation alteration under high-sugar conditions. Our work also suggests that glycosylation-mediated Notch signaling extensively remodels the transcriptomes of tracheal cells and is essential for cell fate determination.

## Results

### The single-cell atlas of *Drosophila* trachea

We performed scRNA-seq to catalog the cell types and identify the transcriptional profiles of the *Drosophila* tracheal cells. The major tracheal branches along with the terminal branches and associated progenitors were dissected from 50 white pupae [0 h after puparium formation (APF)] for the preparation of single-cell suspension (Fig. 1a; Methods). Our scRNA-seq dataset contained the transcriptional profiles of 9615 cells in total, estimated to achieve over 9x coverage of the pupal trachea. Unsupervised clustering with Seurat segregated these transcriptomes into 12 clusters (Supplementary Fig. 1). We proceeded to annotate these clusters (Fig. 1b) by employing both reported and newly identified marker genes (Fig. 1c). All the tracheal branches such as the dorsal trunk (DT), dorsal branch (DB), transverse connective (TC), visceral branch (VB), spiracular branch (SB), lateral trunk (LT), ganglionic branches (GB), air sac primordium (ASP), and progenitor cells (PC) were present in the annotated clusters, as shown in Fig. 1b. A modest quantity of fat body (FB), muscles and neuroendocrine cells (NE) were retained within the preparation (Supplementary Fig. 1). The known markers for individual clusters were visualized. *blistered* (*bs*) that encodes the *Drosophila* homolog of mammalian serum response factor (SRF) was predominantly expressed in tracheal progenitors (Fig. 1d), consistent with a previous report[32]. In accordance with work by Rao and colleagues[19], *spalt* (*sal*) and *knirps* (*kni*), which encode two proteins acting as transcriptional repressors, were markers for cells attributed to DT and DB, respectively (Fig. 1d). To assess the accuracy of clustering, we compared the identified marker genes with results from cell-type-specific transcription profiles. First, a correlation analysis of gene expression showed that the PC cluster displayed the highest similarity with the transcriptomic profile obtained from bulk RNA-seq of progenitor cells[31] (Fig. 1e). Furthermore, the transcriptome of the SB cluster exhibited a significant degree of consistency with previous transcriptional profiling achieved through a microarray strategy[22], as shown in Fig. 1e. These observations validate the precision of the clustering procedure. To further characterize the enriched regulons within these annotated clusters, we performed SCENIC analysis and identified a set of transcription factors with binding motifs significantly enriched in specific clusters. These include Bs in PC, Kni in DB, and Sal in DT (Fig. 1f), representing cluster-specific gene regulatory nodes. Hence, we concluded that this single-cell reference atlas encompassed all the major tracheal cell types and faithfully captured the cellular heterogeneity of the trachea.

To further validate the molecular signatures that are expressed in the domains of the tracheal system (Fig. 2a–c, g–i), we monitored their expression patterns by employing available antibodies, enhancer trap lines, and Gal4 alleles. Sal, a zinc finger transcriptional repressor, was pronounced in DT cells (Fig. 2a, d), which is in agreement with its function in the specification of the dorsal trunk[20,33]. Consistently, the bioinformatic analysis of scRNA-seq suggested that *sal* was expressed in DT and VB, but exhibited a relatively low level in DB or PC (Supplementary Fig. 2). Consistent with a prerequisite role of Kni in dorsal branch morphogenesis[18], the cells constituting DB were marked by *kni*-Gal4-driven nuclear GFP (Fig. 2e). In line with previous reports[19,22],

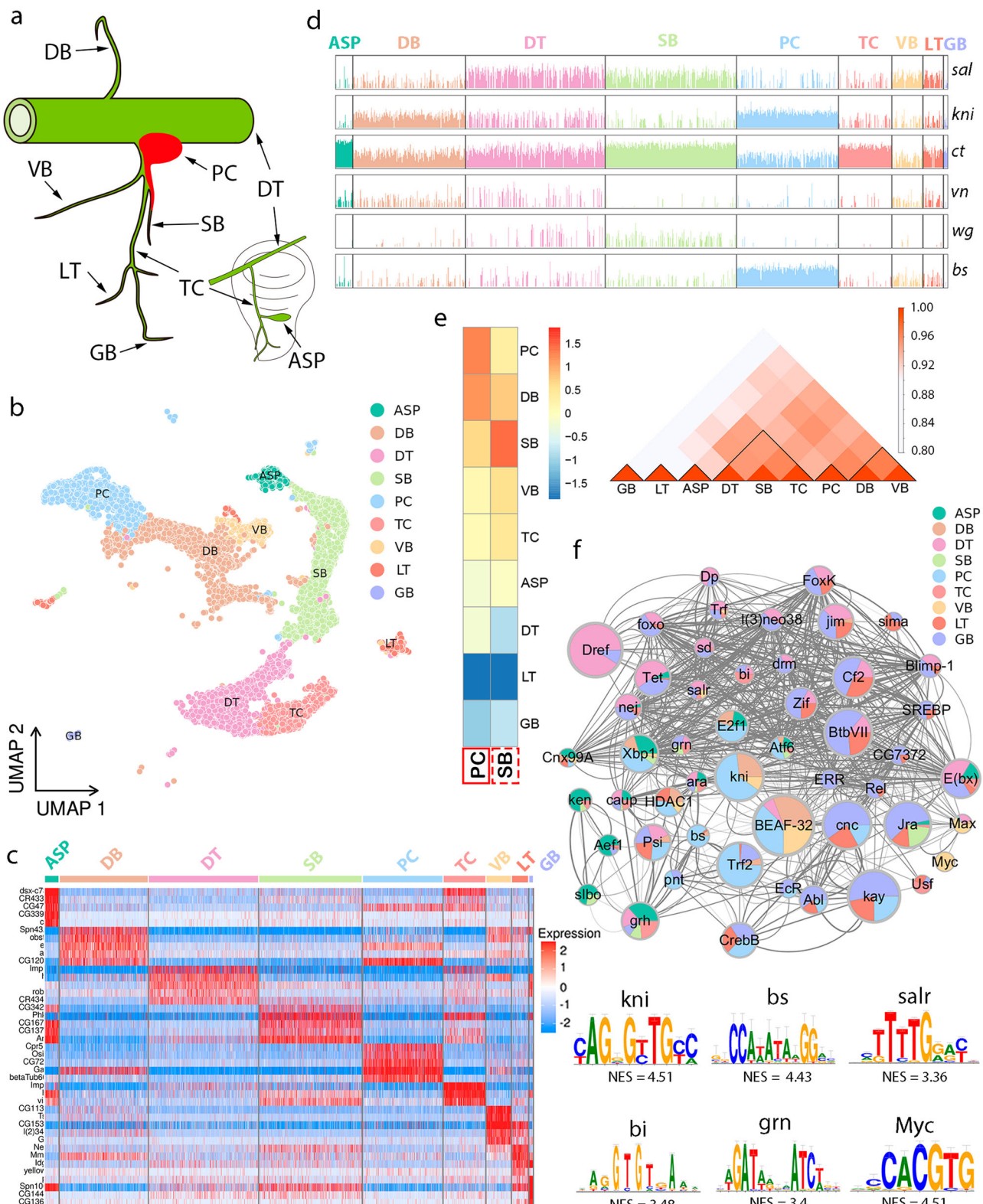

**Fig. 1 | Distinct cell populations in *Drosophila* airway. a** Schematic representation depicting a branched *Drosophila* tracheal system. The bottom right inset depicts the TC, ASP, and an imaginal disc (gray) on the Tr2 metamere. Individual branches are indicated by arrows. **b** UMAP visualization of annotated cell types from *Drosophila* trachea scRNA-seq data. **c** Heatmap showing the expression of the unique genes for each cluster. **d** Bar graphs showing the gene expression levels in individual clusters. **e** Correlation heatmap showing the similarity of bulk RNA-seq data of PCs (red box) and microarray data of SB[22] (dashed box) to the transcriptome of each cluster. RNA-seq data from PCs and microarray data of SBs were queried to the cell types visualized in scRNA-seq. The color scheme indicates the degrees of similarity. Interaction matrices on the right show a correlation between clusters. **f** Cytoscape depicting activity and interaction network of regulons in each cluster. The size of each dot indicates the number of genes directed by the transcription factor. The weight of gray lines connecting dots represents the co-regulated genes. The bottom panels show cis-regulatory elements of indicated transcription factors with normalized enrichment scores (NESs). DT dorsal trunk, DB dorsal branch, TC transverse connective, ASP air sac primordium, VB visceral branch, SB spiracular branch, PC progenitor cells, LT lateral trunk, GB ganglionic branches.

the expression of *cut* (*ct*) was detected in TC and SB (Fig. 2f). In addition, *wg* was exclusively expressed in the domain of SB, as shown in Fig. 2j, which is in fitting with analysis (Supplementary Fig. 2). An EGFR ligand, Vein, was present throughout the VB, but is absent in SB, PC or TC (Fig. 2k). The progenitor cells (PC) were delimited by the expression of the transcription factor, Bs (Fig. 2l). These identified marker genes were co-expressed with other transcription factors in the subclusters (Supplementary Fig. 2), suggesting previously unappreciated heterogeneity within each cell type, an interesting feature that would be further analyzed in following sections. These experimental observations provide corroborative evidence for the markers identified by scRNA-seq. Thus, the single-cell transcriptomes offer insightful indications of specific markers for distinct subpopulations of tracheal cells.

### The diversification of tracheoblasts

The identifiable populations of tracheoblasts that possess property of multipotency are DB and PC[21,34], and they are in active proliferation (Supplementary Fig. 3). The DB cluster of cells was characterized by the notable expression of *exp* and *kni* (Fig. 3a). The dorsal branches extending from the two sides of metameres connect across the dorsal midline (Fig. 3b). In support of the functional significance of the identified DB-specific signature genes, expression of RNAi against *exp* or *kni* resulted in unfused branches and abnormal tip cells in DB, and severely retarded anastomosis formation (Fig. 3c–e). The PC cluster of cells was featured by the expression of *bs*, *mirr*, *bru2*, and *wun* (Fig. 3f). Perturbation of these marker genes by expressing RNAi constructs caused aberrant proliferation and migration of progenitors (Fig. 3g, h). Specifically, whereas expression of *bru2RNAi* increased the proliferation of progenitors, RNAi targeting of *bs*, *mirr* or *wun* reduced their proliferation (Fig. 3g). Furthermore, RNAi directed against *bs* or *bru2* promoted migration of progenitors, but knockdown of *mirr* or *wun* suppressed the migration (Fig. 3h). To probe the apparent heterogeneity within progenitors, we performed FindSubCluster which executes unsupervised identification of subclusters under one primary cluster in Seurat and identified four subclusters for PCs (Fig. 3i). The transcriptomic similarities and functional annotations of these individual subclusters indicated their heterogeneity and distinct biological functions (Supplementary Fig. 4a–d). Specifically, two major subclusters of PCs were closely associated with muscle function (Supplementary Fig. 4a, b), which is in agreement with previous reports on the interdependence between tracheal tubes and muscle[35,36]. Perturbation of muscle in the larval stage significantly reduced the proliferation of PCs (Supplementary Fig. 4f). We also found that the majority of PCs exhibited the expression of *myospheroid* (*mys*), a marker gene functionally related to the muscle (Supplementary Fig. 4h–i'). In addition, the expression of Osi15, another marker gene for these two subclusters of PCs (Fig. 3j), was detected (Supplementary Fig. 4j, k'). We proceeded to analyze the other two notable subclusters: one subcluster was characterized by the expression of *cut* (*ct*) (Fig. 3j and Supplementary Fig. 4c), consistent with the roles of Cut in actin- and microtubule-based cytoskeletal development[37], and the other marked by the enrichment of *ImpL2* and *Thor*, genes functionally associated with insulin-like receptor signaling (Fig. 3j and Supplementary Fig. 4d), which is in fitting with hormonal control of tracheal progenitors by insulin[31]. The *cut*-expressing subpopulation of PCs was verified by an enhancer trap Gal4 line and an antibody against Cut (Fig. 3k, l). Interfering with *ct* expression by RNAi promoted the proliferation and migration of progenitors (Fig. 3g, h). Reciprocally, overexpression of *ct* reduced the migration of progenitors (Fig. 3h). Using a reporter that assays insulin receptor (InR) activity to label the insulin-responsive subpopulation[31], it was found that the *cut*-expressing progenitors displayed low levels of InR activity, suggesting the presence of two distinct progenitor subpopulations characterized by either *cut* expression or InR activity (Fig. 3m-m'''). The *cut*-positive progenitors exhibited

differences from the abovementioned subpopulations that are functionally related to muscle since perturbation of muscles did not alter the number of *cut*-expressing progenitors (Supplementary Fig. 4g).

Additionally, subclustering analysis performed on other cell clusters further revealed their heterogeneity (Supplementary Fig. 5). DT cells were functionally annotated as three subclusters (Supplementary Fig. 5a), among which the DT2 subpopulation was considerably different from the other two subclusters (Supplementary Fig. 6a). Pseudotime trajectory analysis identified developmental trajectories from DT2 towards SB or DT0 populations, suggesting that DT2 displayed multipotency and harbored undifferentiated cellular state compared with other subpopulation (Supplementary Fig. 6b, c). Together, these results document the diversity of tracheoblasts and suggest that the identified populations are functionally driven by specific markers.

### Transformation of specific tracheal branch

The aforementioned results described potential master regulators in DT and DB. We next attempted to explore whether the identified marker genes could endow tracheal cells with lineage-specific characteristics. While the cells in DT expressed the Notch ligand Delta, *serp* showed a high expression in DB, but was inadequately detected in DT (Fig. 4a, b, l). We observed that Delta expression was induced in DB cells when *spalt* (*sal*) was ectopically expressed in these cells, as directed by the DB-specific *kni*-Gal4 (Fig. 4c, m). Correspondingly, these *sal*-expressing DB cells also exhibited morphological resemblance to the polyploid DT cells, namely large and distanced nuclei, suggesting a potential acquisition of DT cell identity (Fig. 4d). Twin of m4 (Tom), a regulator of Notch signaling, is specifically expressed in DT and a subpopulation of SB. Ectopic expression of *Tom* in DB endowed DB cells with the expression of *Delta* (Fig. 4e, f, k, m). Conversely, DT cells that ectopically expressed Kni, the specific marker for DB, adopted an elevated expression of *serp*, which normally serves the molecular identification of DB (Fig. 4g–j, n). These results suggest that cells in different branches are able to transform their cellular identity by forcing the expression of marker genes.

### Notch signaling in tracheal branches

The scRNA-seq results we have presented suggest that various transcription factors and regulons govern the specification of tracheal cells. Further analysis revealed that components in Notch signaling alongside its canonical targets were produced in various tracheal cell types (Fig. 5a). Specifically, we examined target genes of Notch signaling, mainly the components of Enhancer of split complex[38]. The results showed the enriched expression of *E(spl)m3-HLH* and *E(spl)m7-HLH* in the multipotent DB cells (Fig. 5b). Furthermore, Peb and a member of the Bearded family, Tom[39,40], two transcription factors that play opposing roles in Notch signaling, displayed complementary distribution in different tracheal populations, suggesting that Notch signaling plays a critical role in the specification of tracheoblasts (Fig. 5c). Then, we utilized Cellchat[41] to investigate intercellular communication between the diverse tracheal populations and the results of Cellchat implicated that the progenitors received ligands generated by multiple branches (Fig. 5d). To further validate this prediction, we perturbed the Notch ligand, Delta in SB, TC or DT using specific Gal4 alleles. Under these conditions, the activity of Notch signaling in progenitors was compromised, as assessed by Peb staining (Supplementary Fig. 7).

To directly assess Notch signaling activity in tracheoblasts, we assayed the expression of a Notch reporter, NRE-GFP. The reporter has been designed to place the GFP gene *in cis* to a consensus DNA binding site (NRE) of the Suppressor of Hairless [Su(H)], making it a sensitive and robust indicator of Notch signaling[42]. The results showed that Notch signaling activity was obvious in a subset of progenitors as well as at the junction between TC and DT (Fig. 5e, e'). Some of these Notch-

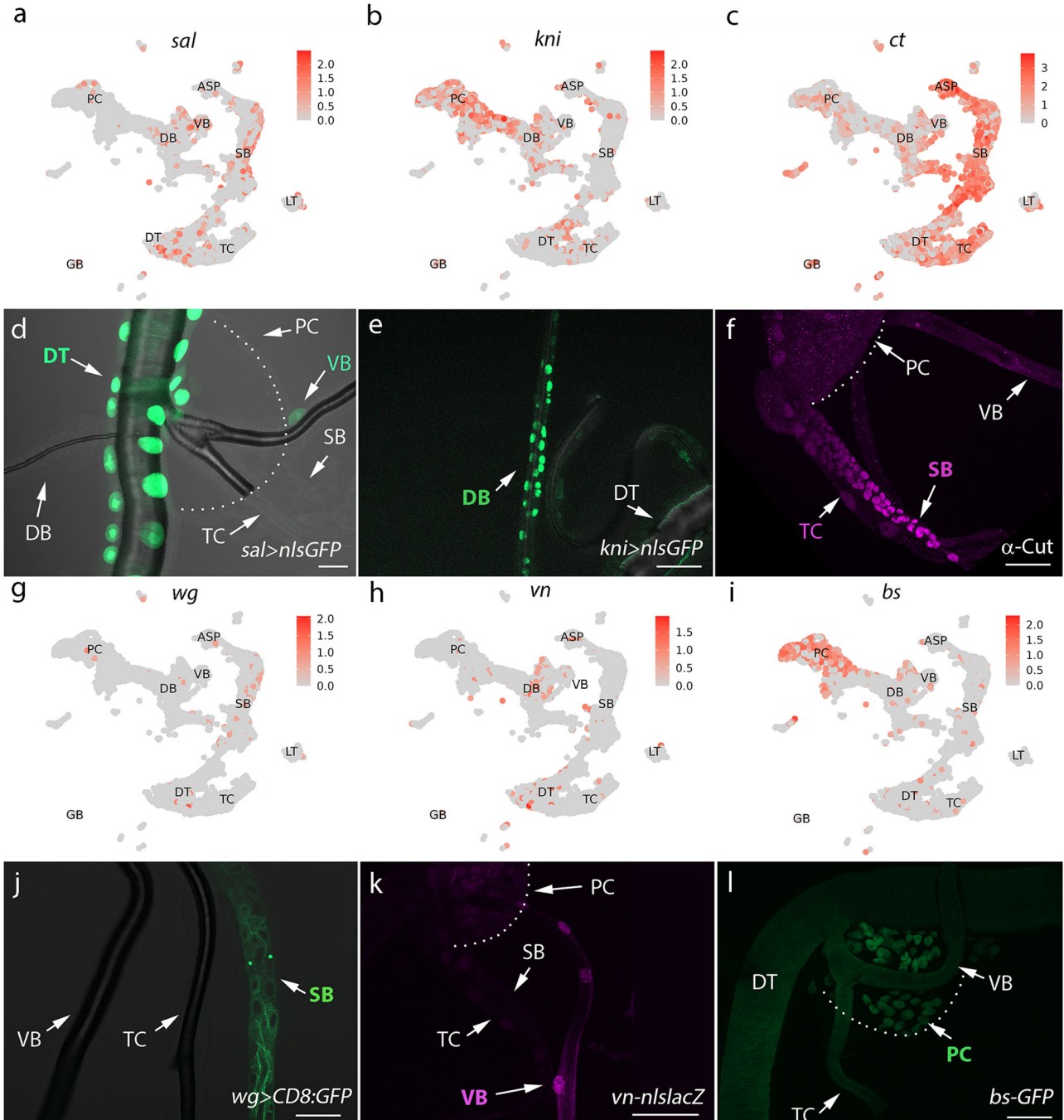

**Fig. 2 | Patterns of gene expression in each cluster of *Drosophila* trachea.**
**a–c, g–i** Expression levels of marker genes in each cluster of tracheal cells. d,e, Confocal images of nls-GFP expression under the control of *sal*-Gal4 (**d**) or *kni*-Gal4 (**e**). **f** Fluorescent image showing *cut*-expressing spiracular branch and transverse connective. The pupal trachea was stained with an antibody against Cut. Arrows point to TC and SB. **j** Confocal image showing the expression of *wg*-Gal4-driven *UAS-CD8:GFP* in SB. **k** The expression of *vein* in VB is visualized by a reporter of *vn*-*nlslacZ* in which a nuclear lacZ is under the control of promoter of *vein* gene. Arrow indicates VB. **i** The expression of Bs-GFP in tracheal progenitors. Scar bars: 20 μm (**d–f, j, l**), 50 μm (**k**). Each experiment was repeated independently with similar results three times (**d–f, j–l**). DT dorsal trunk, DB dorsal branch, TC transverse connective, ASP air sac primordium, VB visceral branch, SB spiracular branch, PC progenitor cells.

responsive progenitors belonged to the aforementioned *cut*-expressing population (Fig. 5f, f').

To further explore the genetic program involved in the diversification of tracheoblasts and understand the function of Notch signaling, we performed RNA-seq of progenitors during their initial process of differentiation. Approximately 10 tracheoblasts from individual progenitor clusters of wandering L3 larvae, white pupae (0 h APF), and 2 h-APF-pupae were dissected[31]. The results of these experiments

revealed that the expression level of *peb* elevated when the differentiation of tracheoblasts commences (Fig. 5g). Several Notch-regulated members, including Enhancer of split m2, Bearded family member (E(spl)m2-BFM) and Enhancer of split m3, helix-loop-helix (E(spl)m3-HLH), also showed increased expression during the differentiation of progenitors (Fig. 5h, i), suggesting that Notch signaling is activated at this stage. Expression of *bs*, a transcription factor that initiates cell specialization was also increased during larval-pupal

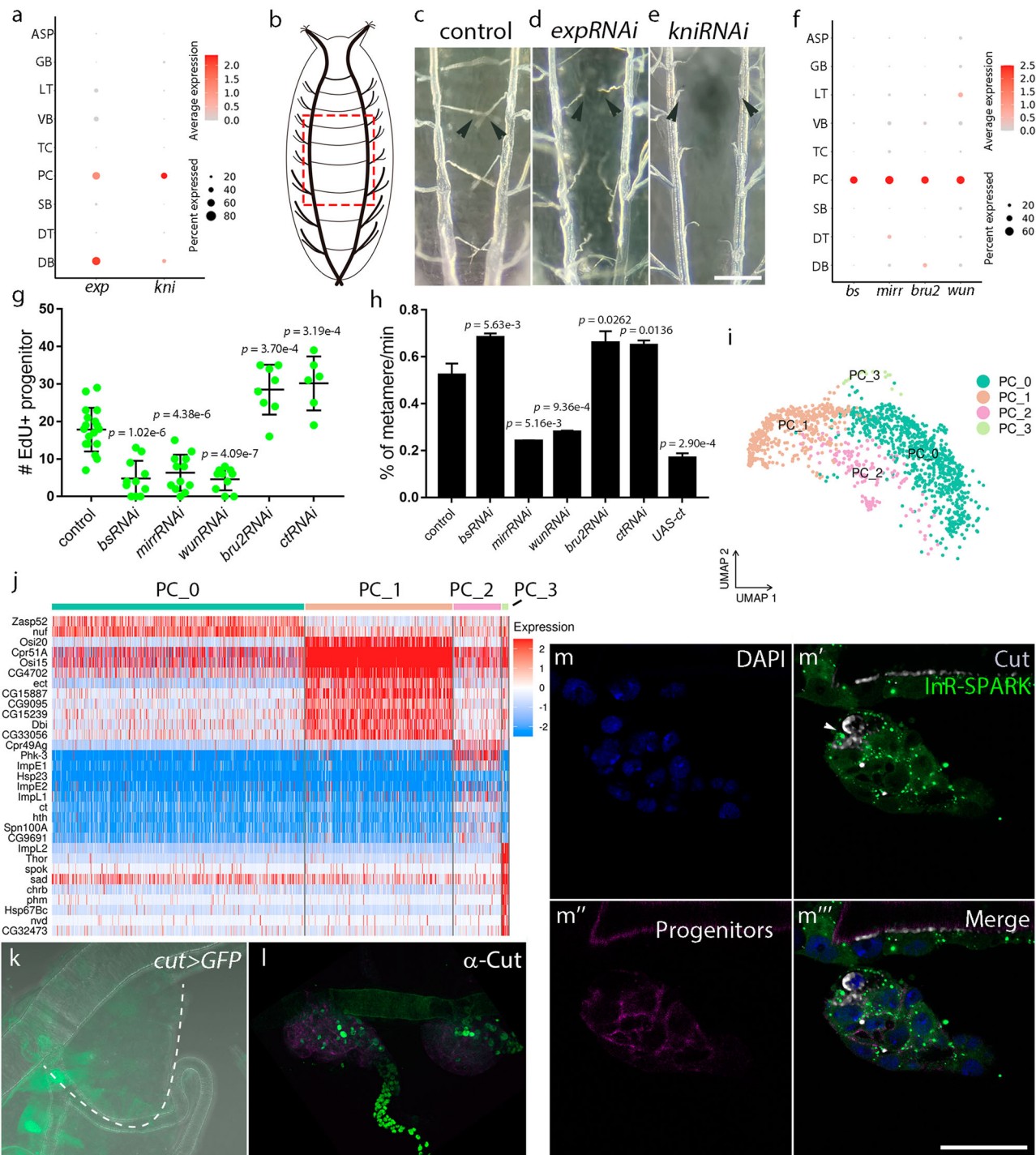

**Fig. 3 | Heterogeneity of progenitors and dorsal branch. a** Dotplot showing the expression of *exp* and *kni* in each cluster. **b** Schematic diagram showing the tracheal system of a pupa. The red boxed region indicates DB. **c–e** Dependence of DB on Exp and Kni. Images of DB in control (left), *expRNAi* (middle) and *kniRNAi* (right) flies. Each experiment was repeated independently for three times with similar results. **f** Dotplot showing the expression of *bs*, *mirr*, *bru2*, and *wun* in each cluster, with a significant enrichment in PC. **g** Scatter plot showing the number of incorporated EdU foci in tracheal progenitors of control (*n* = 18), *bsRNAi* (*n* = 11), *mirrRNAi* (*n* = 12), *wunRNAi* (*n* = 10), *bru2RNAi* (*n* = 8) and *ctRNAi* (*n* = 6) pupae. **h** Bar graph plotting the velocity of migrating progenitors (*n* = 3). **g**, **h** Three biologically independent replicates were performed for each experiment. Data are presented as mean values ± SD. An unpaired two-tailed *t*-test was used for all statistical analyses. No adjustments were made for multiple comparisons. **i** UMAP plot representing subclusters of PC. **j** Heatmap showing the expression of the unique genes for each cluster. **k**, **l** Cut expression in progenitor cells. **k** *cut*-expressing progenitors were visualized by *ct*-Gal4-driven *UAS-GFP* expression. PCs are indicated by a dashed line. **l** Cut expression was through immunostaining of Cut in the dissected trachea of P[B123]-RFP-moe flies. **m–m'''**, The expression of Cut and InR-SPARK in PCs of L3 larvae. **m** Nuclei of progenitors are indicated by DAPI staining. **m'** The presence of *cut*-expressing progenitors and progenitors with high insulin receptor activity as indicated by GFP droplets. Arrowhead points to progenitors expressing Cut while exhibiting low InR activity. **m''** PCs are marked by P[B123]-RFP-moe. **m'''** Merge image. **k–m'''** Each experiment was repeated independently with similar results for three times. DT dorsal trunk, DB dorsal branch, TC transverse connective, ASP air sac primordium, VB visceral branch, SB spiracular branch, PC progenitor cells, LT lateral trunk, GB ganglionic branches. Scar bars: 200 μm (**c–e**), 50 μm (**k**, **l**), 20 μm (**m–m'''**). Source data are provided as a Source Data file.

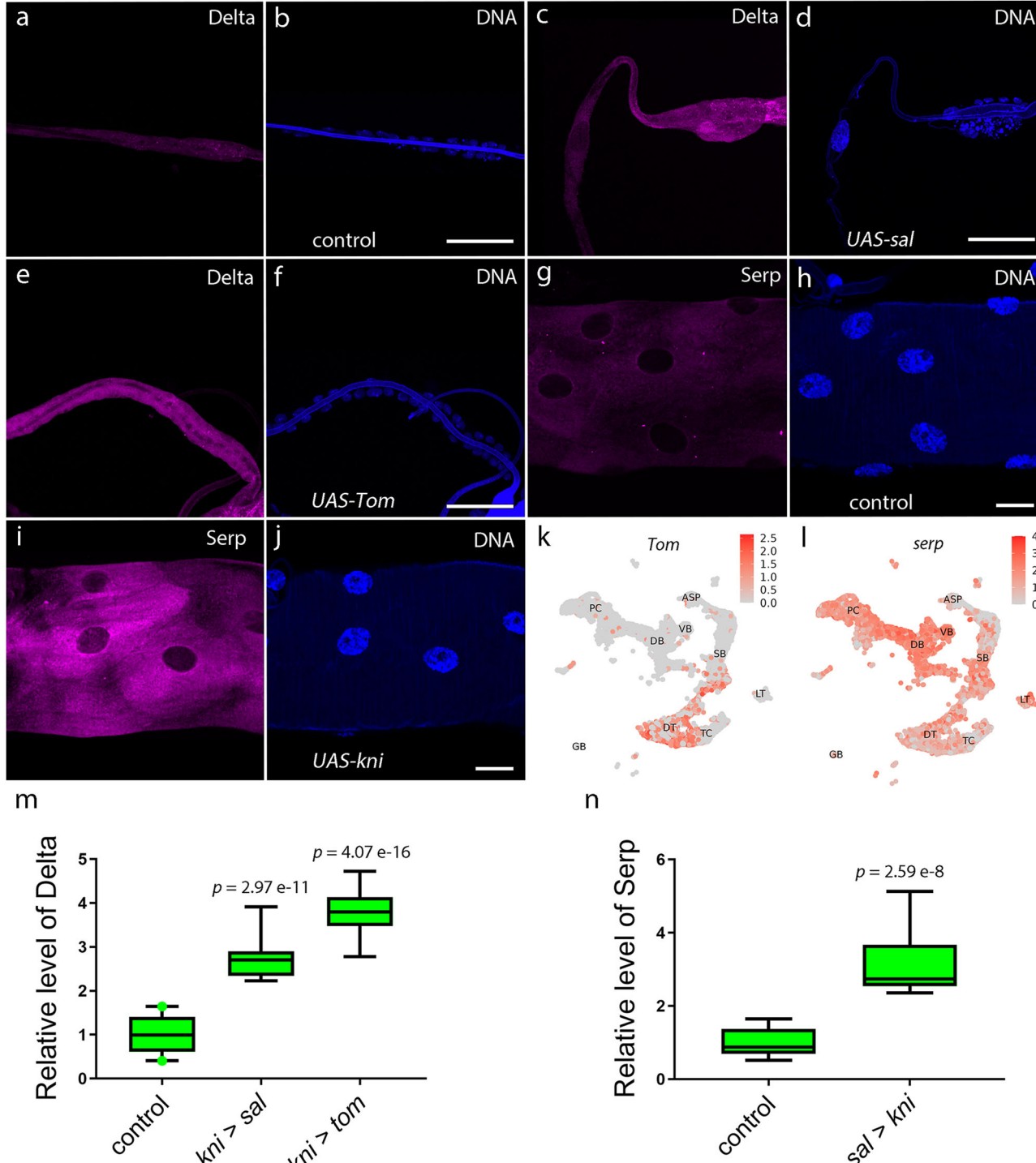

**Fig. 4 | Transcription factors transform dorsal branch and dorsal trunk cells.**
**a–f** Ectopic expression of Spalt or Tom in the dorsal branch activates Delta which is normally expressed in the dorsal trunk. Dorsal branches of control, *UAS-sal*, or *UAS-Tom* flies were stained with an antibody against Delta (**a**, **c**, **e**), and DNA was stained with DAPI (**b**, **d**, **f**). **g–j** Ectopically expressed Kni induces Serpentine (Serp) expression. The dorsal trunk of control or *UAS-kni* flies were stained with an antibody against Serp (**g**, **i**) and DAPI (**h**, **j**). **a–j** Each experiment was repeated independently with similar results for four times. **k**, **l** UMAP plot representing expression of *Tom* (**k**) and *serp* (**l**) in each cluster of tracheal cells. m, Box plot represents Delta levels in control ($n = 20$), *UAS-sal* ($n = 12$) or *UAS-Tom* ($n = 12$) flies. **n** Box plot showing Serp contents in control ($n = 14$) or *UAS-kni* ($n = 12$) flies.

**m**, **n** Data are presented as median with minima and maxima. 25th–75th percentile (box) and 5th–95th percentile (whiskers) as well as outliers are indicated in the box plots. More than four biologically independent experiments were performed. An unpaired two-tailed *t*-test was used for all statistical analyses. No adjustments were made for multiple comparisons. Genotypes: **a**, **b** *kni-Gal4/+*; *tub-Gal80^ts/+*. **c**, **d** *kni-Gal4/UAS-sal*; *tub-Gal80^ts/+*. **e**, **f** *kni-Gal4/+*; *UAS-Tom/tub-Gal80^ts*; *UAS-Tom/+*. **g**, **h** *sal-Gal4/tub-Gal80^ts*. **i**, **j** *UAS-kni/+*; *sal-Gal4/tub-Gal80^ts*. DT dorsal trunk, DB dorsal branch, TC transverse connective, ASP air sac primordium, VB visceral branch, SB spiracular branch, PC progenitor cells. Scar bars: 50 μm (**a–f**), 20 μm (**g–j**). Source data are provided as a Source Data file.

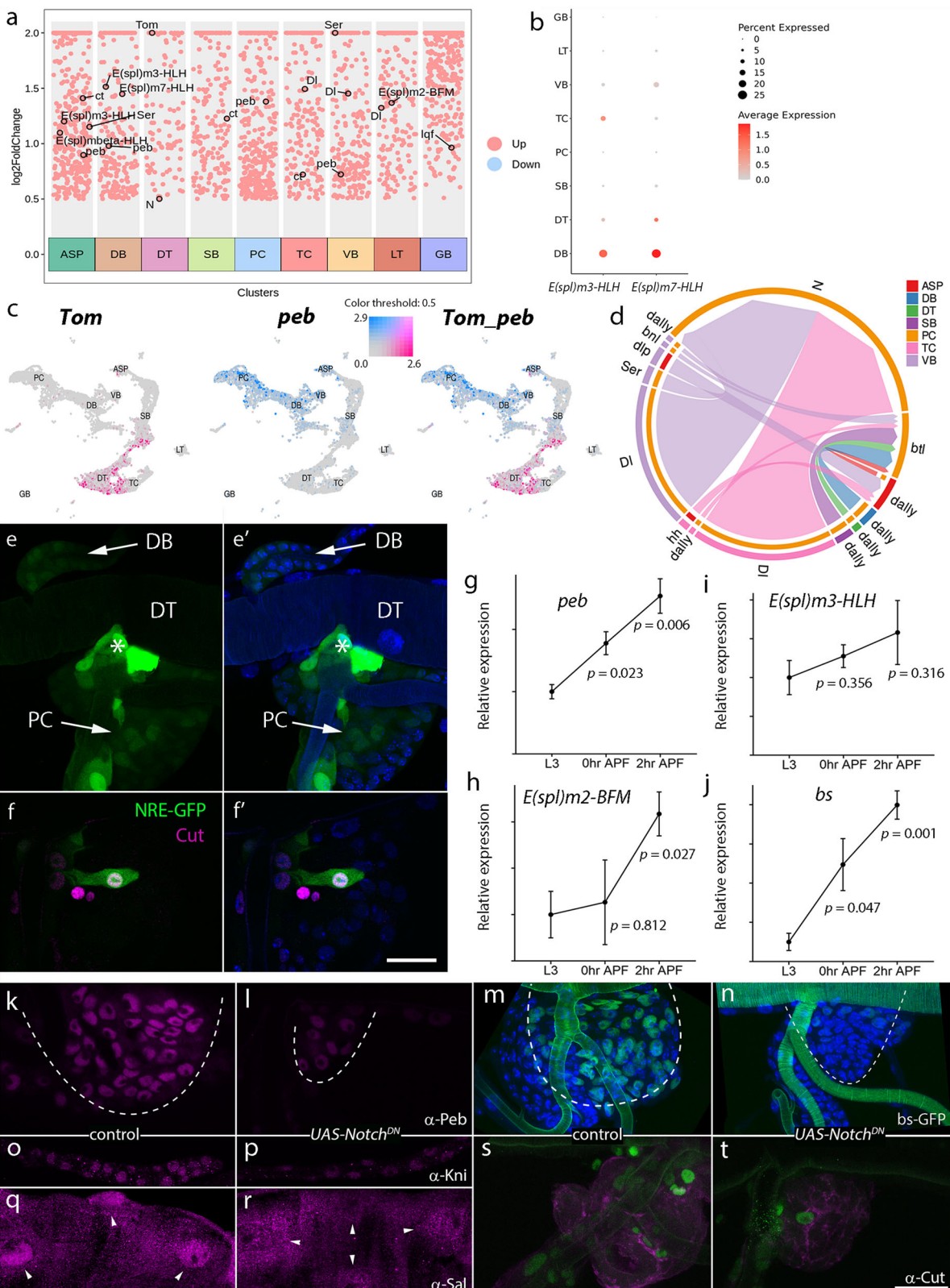

transition, as shown in Fig. 5j. To further determine the effects of Notch signaling on these regulators, we perturbed Notch signaling by expressing a dominant negative form of Notch, Notch[DN43]. In comparison with controls, the levels of the transcription factors Peb and Bs were reduced in tracheal progenitors (Fig. 5k–n). Meanwhile, the expression of Kni in the dorsal branch and Sal in the dorsal trunk were both decreased by the reduction of Notch signaling (Fig. 5o–r). In addition, Cut expression in tracheal progenitors was attenuated upon the perturbation of Notch (Fig. 5s, t). Taken together, these observations suggest that Notch signaling plays an integral role in the process of tracheal cell specification and modulates multiple cell fate determinants.

**Fig. 5 | Tracheal morphogenesis depends on Notch signaling. a** Scatter plot depicting the expression of the Notch module as well as targets in tracheal cells. **b** Dotplot showing the expression level of Notch targets, *E(spl)m3-HLH* and *E(spl) m7-HLH*, in each cluster. **c** The expression of *Tom* and *peb* in each cluster of tracheal cells. **d** Chord diagram depicting intercellular communication between tracheal clusters. **e, f'**, The expression of Notch reporter, NRE-GFP, in the trachea (**e**). The trachea of NRE-GFP fly was stained with DAPI (**e'**). The arrows point to PCs or DB. Asterisks denote the junction between TC and DT. **f-f'**, The expression of NRE-GFP reporter and Cut in progenitors. The trachea of NRE-GFP flies were stained with antibodies against Cut and DAPI. **f'** Merge image. **g–j** Relative expression levels of *peb* (**g**), *E(spl)m2-BFM* (**h**), *E(spl)m3-HLH* (**i**) and *bs* (**j**) in PCs at indicated developmental stages. Gene expression values are normalized to that of L3. *n* = 3 for each genotype. *p*-value, 0 h APF: *peb* (0.023), *E(spl)m2-BFM* (0.812), *E(spl)m3-HLH* (0.356) and *bs* (0.047); 2 h APF: *peb* (0.006), *E(spl)m2-BFM* (0.027), *E(spl)m3-HLH* (0.316)

and *bs* (0.001). Three biologically independent experiments were performed. Scatter plots show the upregulation of indicated genes during pupariation. Data are presented as mean values ± SD. An unpaired two-tailed *t*-test was used for all statistical analyses. No adjustments were made for multiple comparisons. **k–r** Dependence of transcription factors on Notch signaling in the trachea. α-Peb antibody immunostaining (**k, l**), bs:GFP expression (**m, n**), α-Kni antibody immunostaining (**o, p**) and α-Sal antibody immunostaining (**q, r**) in the PCs of control and *Notch^DN* flies. Arrowheads point to nuclei (**q, r**). **s, t** α-Cut antibody immunostaining of PCs labeled by P[B123]-RFP-moe in control and *Notch^DN* flies. Each experiment was repeated independently with similar results for three times (**e, f', k–t**). DT dorsal trunk, DB dorsal branch, TC transverse connective, ASP air sac primordium, VB visceral branch, SB spiracular branch, PC progenitor cells, LT lateral trunk, GB ganglionic branches. Scar bars: 30 μm (**e, f', k–t**). Source data are provided as a Source Data file.

## Glycosylation in the trachea upon high-sugar diet

Gene Ontology (GO) analysis of the differentially expressed genes (DEGs) during the activation of tracheal progenitors revealed a significant enrichment of protein glycosylation-related terms (Fig. 6a). Molecules involved in glycosylation were elevated during the activation of tracheal progenitors (Fig. 6b). We surveyed genes essential for *N*-acetylglucosamine (GlcNAc) modification in the cytoplasm and nucleus by OGT or in EGF repeating domain by EOGT or *N*-acetylgalactosamine (GalNAc) modification[44,45]. The majority of them were confirmed to be active in different cell types (Fig. 6c–e). To better understand the dynamics of this modification in the trachea, we examined the expression levels of key enzymes contributing to glycosylation in the tracheal progenitors (Fig. 6f). The results showed that the levels of *mummy* (*mmy*) encoding a UDP-N-acetylglucosamine diphosphorylase, *fringe* (*fng*) that encodes a β−1,3-N-acetylglucosaminyltransferase that modifies EGF domains, and *kuduk* (*kud*) that encodes a subunit of the oligosaccharyltransferase (OST) complex (Fig. 6e), were increased during the larval-pupal transition when tracheal progenitors are activated and differentiate, suggesting that protein glycosylation is elevated during this developmental stage (Fig. 6f). Given the concurrent activity of Notch signaling at this stage (Fig. 5g–i) and the possible modification of Notch receptors by *O*-linked glycosylation[46,47], we hypothesized that protein glycosylation contributed to activity of Notch signaling. To this end, we perturbed protein *O*-glycosylation by expressing RNAi against the aforementioned glycosylation enzymes. RNA-seq analysis of *mmyRNAi* flies identified the Notch signaling pathway as one of the prominent functional classes among the differentially expressed genes (DEGs) (Supplementary Fig. 8a). The downstream genes of Notch signaling, such as *E(spl)mβ-HLH*, *E(spl)m2-BFM* and *E(spl)m7-HLH*, were reduced by the expression of *mmyRNAi* (Fig. 6g–i). Meanwhile, other signaling pathways such as insulin, Hippo, or Dpp/TGF-β were not affected by *mmy* abrogation, as assayed by specific reporters of these signaling (Supplementary Fig. 8b–j). Furthermore, upon the expression of *mmyRNAi*, Notch signaling activity, as indicated by NRE-GFP, was severely diminished in PC, DB, and the junction between DT and TC (Fig. 6j–m' and Supplementary Fig. 9a–d). Additionally, RNAi-induced depletion of *O-fut1* or *fng*, genes encoding the enzymes catalyzing protein *O*-fucosylation and subsequent GlcNAc modification respectively, reduced the expression of NRE-GFP reporter in both PC and DB cells (Fig. 6n–q' and Supplementary Fig. 9a–d). These results support the notion that glycosylation promotes the activity of Notch signaling.

Since protein glycosylation is altered in hyperglycemia or diabetic conditions[48], we sought to determine whether a high-sugar diet influenced protein glycosylation by raising larvae on a normal diet (ND) or sucrose-enriched high-sugar diet (HSD). Compared with ND control, the activity of insulin signaling in the trachea was increased in the presence of HSD, as assessed by InR-SPARK and Akt-SPARK reporters[31], suggesting that tracheal cells are competent for responding to high-

sugar conditions (Supplementary Fig. 10). Next, we examined the expression levels of the aforementioned enzymes such as *mmy*, *fng*, and *kud* in our scRNA-seq data, and observed that they were concurrently up-regulated in HSD, which further supports the notion that glycosylation modification is enhanced by the high-sugar condition (Fig. 6r). Additionally, HSD also promotes the expression of *pgant35A* (Fig. 6r), which encodes a UDP-GalNAc:polypeptide *N*-acetylgalactosaminyltransferase, is required for tracheal tube formation[49] and is essential for *Drosophila* viability[50]. To probe a direct interaction between glycosylation and Notch receptors, we adopted the Click-it GlcNAc enzymatic labeling system that recognizes GlcNAcylation modification on its target proteins. The results of this experiment showed that the GlcNAc-bearing fraction of the Notch protein was significantly increased in the presence of HSD (Fig. 6s). Meanwhile, the analysis of tracheal protein extracts showed that the gross abundance and the number of *O*-GlcNAc-modified proteins were slightly increased upon HSD treatment (Supplementary Fig. 11a). Specifically, elevated level of *O*-GlcNAc-associated Notch receptors was observed in HSD group (Supplementary Fig. 11b). In aggregate, these results suggest that HSD promotes protein glycosylation and potentiates the enzymes involved in glycosyl biosynthetic and catalytic pathways.

## Notch-dependent differentiation trajectory of tracheal cells

The data described thus far suggest that Notch signaling in tracheal cells is fine-tuned by post-translational modification such as glycosylation and impacts a variety of effectors in different cell clusters. In a developmental trajectory of DB, DT, and PC cells visualized by plotting gene expression across pseudotime, a slew of Notch targets as well as genes related to glycosylation were expressed in early PC and DB cells, but diminished over pseudotime, suggesting a dynamic regulation of Notch signaling and glycosylation along with cell lineage of tracheoblasts (Fig. 7a). The expression of *serrate* (*ser*) and *Delta* (*Dl*) peaked in DT cells (Fig. 7a), which is consistent with a previous analysis of gene expression of DT[19]. We examined the expression of cell type-enriched marker genes that depended on Notch signaling (Fig. 5k–p) and observed that the expression levels of *bs*, which is enriched in progenitor cells, *kni*, the marker gene of DB cells, and *peb* concurrently vanished from progenitors to differentiated tracheal cells (Fig. 7b–d). These observations further support the role of Notch signaling in the differentiation of tracheoblasts.

To investigate the cellular transition and biological progress of tracheoblasts, we utilized several analytical tools to visualize the developmental trajectories with our scRNA-seq data. RNA velocity, a representation of cellular state shifting, illustrated that the PC and DB clusters were able to progress towards differentiation cell types (Fig. 7e and Supplementary Fig. 12a)[51]. Consistently, trajectory inference generated by partition-based graph abstraction (PAGA)[52] indicated that the PC and DB cells possessed an undifferentiated cellular

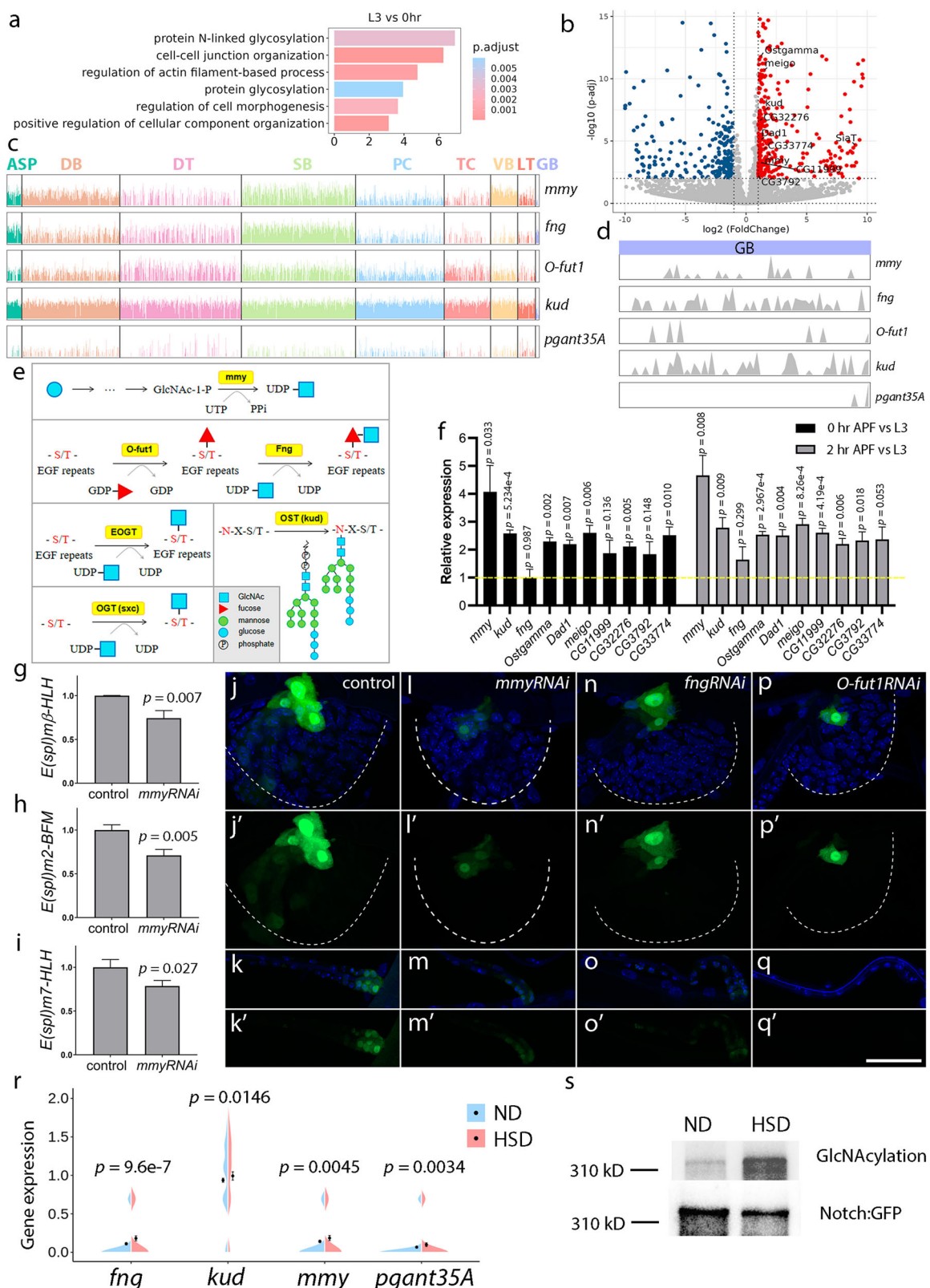

state, compared with DT cells (Supplementary Fig. 12b–e). In accordance with these results, CytoTRACE [Cellular (Cyto) Trajectory Reconstruction Analysis using gene Counts and Expression], a framework that computationally predicts the differentiation status[53], indicated a general gain of differentiation property from PC, DB to DT, TC (Fig. 7f). To test whether HSD impacts cell differentiation, we generated a scRNA-seq dataset of the trachea from HSD-treated flies (HSD

group) in addition to the standard food-reared flies (ND group). Both datasets overlap for all the clusters containing cells from each condition (Fig. 7g, h). CytoTRACE analysis indicated that the differentiation of tracheal cells was severely perturbed upon HSD treatment (Fig. 7i). Under HSD conditions, a substantial amount of differentiated tracheal cells was observed in the PC cluster, while the differentiation status of DT cells was reduced, as shown in Fig. 7i.

**Fig. 6 | HSD affects protein glycosylation. a** Bar graph depicting functional clusters among the differentially expressed genes (DEGs). Hypergeometric test (one-sided) and False Discovery Rate (FDR) adjustment were used. **b** Volcano plot by DESeq2 in Wald test with FDR showing the DEGs (red: up-regulated; blue: down-regulated). **c, d** Track plot showing expression levels in each cluster (**c**) and GB (**d**). **e** Schematic cartoon represents an overview of biosynthetic and catalytic reactions of GlcNAcylation. **f** Expression of genes in protein glycosylation in PCs of 0 h APF (black) and 2 h APF (gray) relative to that of L3. Dashed line indicates levels in L3 set as 1. *p*-value, 0 h APF: *mmy* (0.033), *kud* (5.234e-4), *fng* (0.987), *Ostgamma* (0.002), *Dad1* (0.007), *meigo* (0.006), *CG11999* (0.136), *CG32276* (0.005), *CG3792* (0.148), *mmyCG33774* (0.010); 2 h APF: *mmy* (0.008), *kud* (0.009), *fng* (0.299), *Ostgamma* (2.967e-4), *Dad1* (0.004), *meigo* (8.256e-4), *CG11999* (4.187e-4), *CG32276* (0.006), *CG3792* (0.018), *CG33774* (0.053). **g–i** Relative levels of *E(spl)mβ-HLH* (**g**), *E(spl)m2-BFM* (**h**) and *E(spl)mβ-HLH* (**i**). **f–i** *n* = 3 per genotype. Data are presented as mean values ± SD. An unpaired two-tailed *t*-test was used for all statistical analyses. No adjustments were made for multiple comparisons. **j–q′** Dependence of Notch signaling on protein glycosylation. **r, s** Protein glycosylation in response to HSD. **r** Split violin plot showing the levels of *fng*, *kud*, *mmy*, and *pgant35A* in PCs in ND (blue) and HSD (red). Data are presented as mean values ± SD. *n* = 2008 cells examined over three independent experiments. A two-sided Wilcoxon rank sum test with Bonferroni correction was used. **s** Glycosylation of Notch in ND and HSD-fed flies. Each experiment was repeated independently with similar results three times (**f–q′**, **s**). DT dorsal trunk, DB dorsal branch, TC transverse connective, ASP air sac primordium, VB visceral branch, SB spiracular branch, PC progenitor cells, LT lateral trunk, GB ganglionic branches, ND normal diet, HSD high-sugar diet, UTP Uridine 5′-triphosphate, UDP-GlcNAc N-acetylglucosamine, GDP Guanidine 5′-diphosphate, EOGT EGF-domain O-GlcNAc transferase, OGT *O*-GlcNAc transferase, OST Oligosaccharyltransferase, S/T serine/threonine, N asparagine. Scar bars: 50 μm (**j–q′**). Source data are provided as a Source Data file.

Next, we employed Gal4 technique for real-time and clonal expression (G-TRACE) system[54] to explore the lineage potential of the PC and DB populations. G-TRACE labels the Gal4-expressing cells and the descendants in red and green fluorescence respectively. The results showed that *kni*-expressing tracheal cells populated the dorsal branch (Fig. 7j), that *cut*-expressing progenitors contributed to the PC domain (Fig. 7k), and that the progenitors expressing *esg* which promotes expression of ImpL2 also constituted the PC population (Figs. 7l and 3j)[55]. In Fig. 7k, l, the GFP-negative progenitors were neither *cut*-expressing nor *esg*-expressing descendants, suggesting the heterogeneity and distinct trajectories of tracheal progenitors. In concordance with above bioinformatic analyses, the descendants from *kni*-expressing DB cells were observed in DT domain in the presence of HSD, suggesting occurrence of considerable cell conversion (Fig. 7m). Furthermore, G-TRACE analysis of HSD-fed flies revealed an expansion of *cut*- and *esg*-expressing progenitors and progeny in PC domain, compared with those in ND, suggesting that cell fate and/or identity of progenitors are influenced by HSD (Fig. 7n, o). In order to further investigate the biological consequences of HSD, we examined differentially expressed genes (DEGs) in all the cell clusters upon HSD. The Notch targets, components of enhancer of split complex, E(spl), were evident in the DEGs, suggesting that Notch signaling is significantly altered in response to HSD (Fig. 7p). Furthermore, GO enrichment analysis indicated significant enrichment of functional clusters associated with Notch pathway (Fig. 7q). Collectively, these results suggest that the differentiation of tracheal cells is influenced by HSD which modulates Notch signaling.

### Transcriptional responses to high-sugar diet

In addition to the progenitors that receive Notch signaling inputs from other branches (Fig. 5d, g–j), Notch signal transduction is also pronounced across multiple tracheal clusters (Fig. 8a, b), which is in agreement with previous analysis of ASP and DB cells[19,35]. In line with a previous report showing FGF-dependent progenitor migration[32], FGFR signaling is preferentially active in the PC population (Fig. 8c, d). Given that the extracellular region of these receptors is commonly glycosylated and that glycosylation of Notch is induced by HSD (Fig. 6n, o), it is tempting to explore the potential cellular responses of tracheal cells to HSD. In this aim, we analyzed the regulons in response to HSD. The results showed that the transcription factors, *bs* and *kni*, which are markers for PC and DB and are dependent upon Notch signaling (Fig. 5m–p), were aberrantly active in other cell populations (Fig. 8e), which is in keeping with the aforementioned observation that *kni*-expressing cells and their progeny emerged in DT domain in response to HSD (Fig. 7m). To further determine whether Notch is activated in response to HSD, we again employed the NRE-GFP reporter and found that the expression of NRE-GFP was notably elevated in DB and PC domains in HSD-treated flies (Fig. 8f–i). Moreover, the number of DB cells and progenitors having Notch activity was increased in response to HSD, suggesting that Notch signaling is enhanced (Fig. 8f–i).

To further investigate the alteration of cell identities under HSD conditions, we projected the scRNA-seq data from the HSD group to the clusters of the single-cell transcriptome of the ND group using the scmap method[56]. The results indicated a considerable amount of cell conversion in the three clusters (DB, PC, and DT) of Notch-responsive cells upon HSD (Fig. 8j), suggesting that the HSD-induced Notch signaling perturbation may lead to cell conversion. Subsequently, we examined additional ligands and cytokines that mediate intercellular communication and found an overall increase in the expression of the ligands across tracheal cells under HSD conditions (Fig. 8k). These results collectively suggest that HSD promotes both juxtacrine and paracrine signaling in the trachea.

## Discussion

This study represents our endeavor to accomplish a comprehensive transcriptome profiling of the *Drosophila* tracheal system which enables us to visualize this developmental program at a molecular level. Although the tracheal cells are perceived as post-mitotic and less heterogeneous, single-cell RNA-seq is able to discern and categorize the distinct populations in the airway. The accessibility to the transcriptome of individual cells even reveals heterogeneity within and pinpoints master regulators responsible for the specification of individual populations. Innovated bioinformatic analyses further resolve the cell fate plasticity of tracheoblasts and the influence of external stimuli such as metabolic status.

Our study reveals heterogeneity within particular stem cell populations which would be a reasonable prediction for their subsequent lineage trajectory. Of the nine clusters corresponding to different branches and cell populations, DB and PC are able to further differentiate into other cell types. The DB cells are capable of differentiation during the pupal stage and generate elaborate and morphologically distinct structures, such as multicellular stalks (MS), unicellular stalks (US), and coiled tracheolar (CT)[34]. Consistent with these histological observations, our scRNA-seq datasets also reveal the multipotency and heterogeneity of DB and PC cells (Fig. 3). The functional diversity of the transcription profile of subpopulations could account for distinct cell fates. For instance, Cut- and ImpL2-positive progenitors although relatively less abundant were identified in silico. The revealed molecular signatures echo their roles in various contexts of stem cells. Escargot (Esg), a member of the Snail family of transcription factors, which potentiates the expression of the insulin antagonist, ImpL2, is responsible for stem cell maintenance[55,57]. The homeodomain transcription factor Cut is required for the identity of renal progenitors[58,59].

Ectopic expression of master regulators causes reprogramming of tracheal cells (Fig. 4), which is consistent with previous reports that Sal and Kni exhibit mutual inhibition during tracheal morphogenesis[60] and

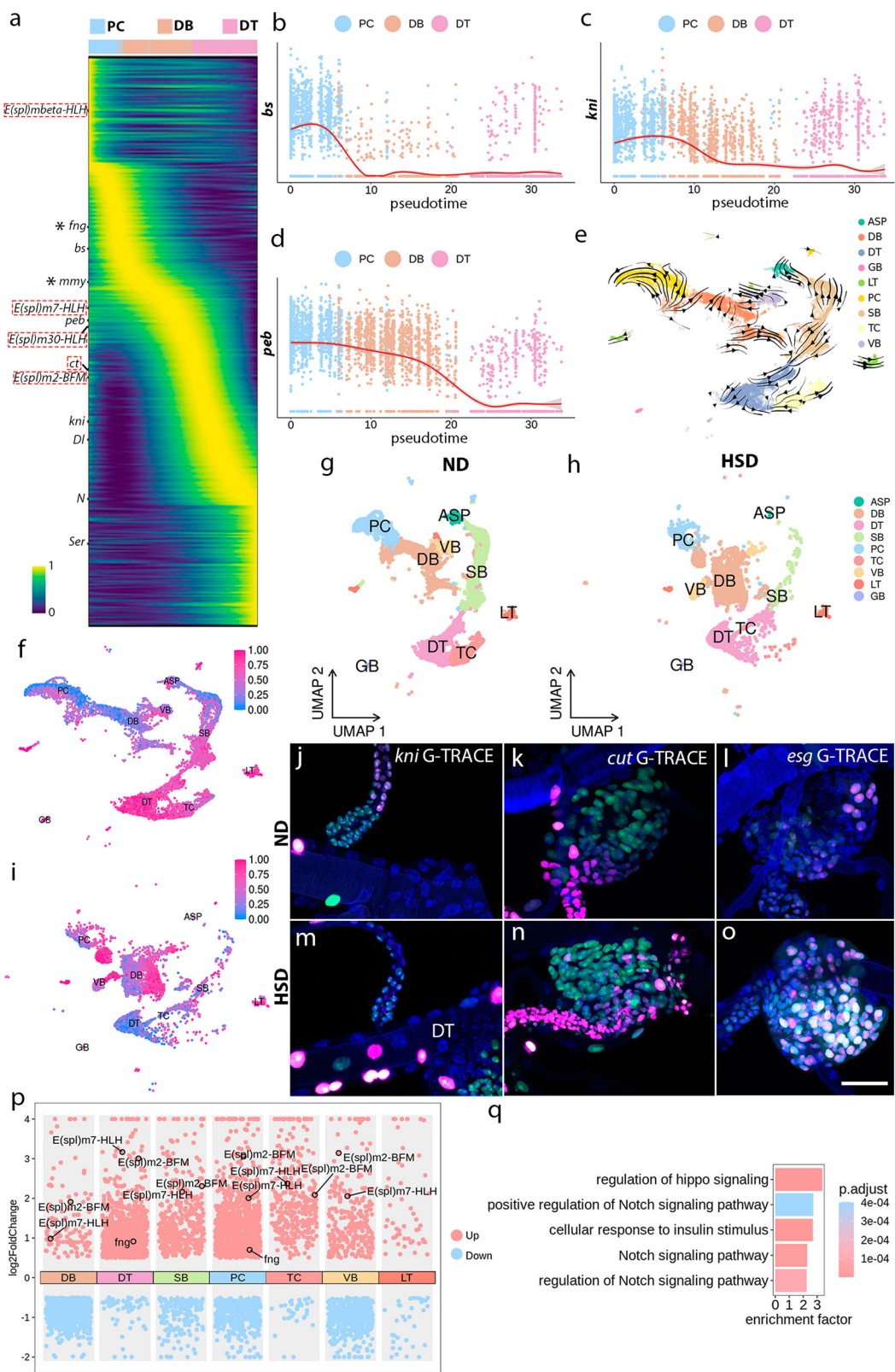

that persistent expression of Sal throughout the trachea renders the tracheal cells DT-like behavior[61]. In addition to the abovementioned transcription factor Bs, as reported previously[32,62], the present study identified factors, including the transcription factor Mirr, the RNA binding protein Bru2, and the phosphatase Wun, that represent additional pivotal regulators in progenitor cells. These genes as well as those markers for subclusters of PC are unlikely to be captured by bulk

RNA-seq even at higher sequencing depth, which has led to under-appreciation of heterogeneity within tracheoblasts. This study points to the important functions of these genes in tracheal progenitors and arouses further investigation of their implications in other aspects of stem cells.

Progenitor cells migrate along the dorsal branch and target the tracheal inducer Bnl/FGF that emanates from the decaying DT cells[32].

**Fig. 7 | The diversification of tracheal cells. a** Heatmap showing gene expression over pseudotime for PC, DB, and DT cells. Dashed boxes denote the Notch targets, components of enhancer of split complex, and *cut* (*ct*). Asterisks point to genes participating in protein glycosylation. **b–d** Representative traces showing the expression patterns of Notch-responsive genes in PC, DB, and DT cells. The expression levels of *bs* (**b**), *kni* (**c**), *peb* (**d**) decreased from multipotent PC and DB cells to terminal differentiated DT cells. **e** Cell maturation map of tracheoblasts estimated by RNA velocity. **f, i** Cytotrace plots showing the state of differentiation of tracheal cells from ND-fed (**f**) or HSD-fed (**i**) flies. **g, h** UMAP visualization of tracheal cell types in ND (**g**) or HSD (**h**). **j–l** Lineage-tracing analyses of *kni*-expressing DB cells (**j**), Cut- (**k**), or Esg-positive (**l**) progenitors. *kni*-Gal4, *ct*-Gal4, or *esg*-Gal4-driven G-TRACE flies were fed on ND. **k, l** Cells that are neither red nor green are not *cut*- or *esg*-expressing progenitor origin. DAPI (blue). **m–o** Lineage-

tracing analysis of *kni*-expressing DB cells (**m**), Cut- (**n**) or Esg-positive (**o**) progenitors. *kni*-Gal4, *ct*-Gal4, or *esg*-Gal4-driven G-TRACE flies were exposed to HSD. DAPI (blue). **j–o** Each experiment was repeated independently with similar results for three times. **p** Scatter plot representing the differentially expressed genes in tracheal cells exposed to ND or HSD. **q** Bar graph depicting functional clusters among the differentially expressed genes between ND and HSD groups. Hypergeometric test and False Discovery Rate (FDR) adjustment were used. Genotypes: **j, m** *kni-Gal4/+;UAS-G-trace/+*. **k, n** *ct-Gal4/UAS-G-trace*. **l, o** *esg-Gal4/UAS-G-trace*. DT dorsal trunk, DB dorsal branch, TC transverse connective, ASP air sac primordium, VB visceral branch, SB spiracular branch, PC progenitor cells, LT lateral trunk, GB ganglionic branches, ND normal diet, HSD high-sugar diet. Scar bars: 50 μm (**j–o**).

In addition to FGF signaling that is captured by scRNA-seq (Fig. 8), our study reveals unappreciated roles of Notch signaling in coordinating gene expression and cell differentiation in distinct tracheal cell populations. Cell-cell signaling mediated by the evolutionarily conserved Notch pathway controls many developmental processes in metazoans[63], and such intercellular communication is also prominent in the fly trachea (Fig. 5). Notch signaling is one of the principal pathways responsible for selecting cell fates within groups of equivalent cells of tracheal branches[23]. The selective activation of Notch signaling acts in concert with a series of transcription factors to determine cell fate during tips of the outer proliferation center (tOPC) neurogenesis[64]. We discovered a graded Notch activity that extends from the multipotent cells to the terminally differentiated population, suggesting a functional relationship between Notch activity and differentiation of tracheal cells (Fig. 5). Progenitors and DB cells share similar transcriptional signatures and are juxtaposed in a developmental trajectory inferred by pseudotime registration. Both *kni* and *bs*, the marker genes of DB and PC respectively, are regulated by Notch signaling (Fig. 5m–p). The PC and DT markers functionally dependent on Notch in the subsequent pseudotime stages align well with expectations from genetic experiments (Fig. 7).

Aberrancy in Notch signaling is implicated in several diseases in human[65], including atherosclerosis[66], cardiac hypertrophy[67], and various types of cancer[68]. It has been shown that high glucose level activates Notch in both mouse models and hyperglycemic patients. The Dll4-Notch1 loop induced by hyperglycemia impairs wound healing[69]. High sugar triggers protein *O*-glycosylation and leads to the accumulation of glycosylation end-products in various contexts[70,71]. We found that a high-sugar diet potentiates Notch signaling and perturbs the diversification of tracheal branches, which is in accordance with the report that GlcNAcylation of Notch receptor promotes Notch-Delta interaction[72]. Glycosylation preferentially impacts proteins on the membrane surface such as ligands and receptors. Notch receptors possess multiple epidermal growth factor-like (EGF) repeats in the extracellular domain that serve as sites for modification by *O*-linked glycans[73,74]. An *N*-acetylglucosamine (GlcNAc) residue can be attached to *O*-linked glycans on Notch by the β−1,3-N-acetylglucosaminyltransferase enzymes such as the Fringe (Fng) family[46,47]. Our results support the notion that the pulse of Notch activity in the trachea is mediated by glycosylation. Expression of several enzymes that contribute to protein glycosylation is modulated by glucose metabolism-regulated Yki signaling[75] during larval-pupal transition (Supplementary Fig. 13). The regulation of Notch activity by glycosylation reinforces the functional interplay between glucose catabolism and cellular signaling and implicates metabolic reprogramming in Notch-dependent cell differentiation and specification.

In summary, the present study provides a resource that uncovers the cell type-specific molecular signatures and gene functions in the fly airway; it further provides mechanistic insights into Notch-dependent tubulogenesis as well as the pathological outcome of HSD in tubular organs. The glycosylation of the Notch receptor may induce an

instructive perturbation of the transcription programs that underlie many metabolic diseases.

## Methods

### Fly husbandry
Flies were cultured on standard cornmeal and agar medium supplemented with standard (0.15 M sucrose) or a high amount of sugar (1 M sucrose)[76]. The culture temperature was 25 °C unless otherwise noted. Detailed information on strains used in this study is listed in Supplementary Table 1. Recipes for diets used are available on request.

### Immunostaining
Immunostaining of *Drosophila* trachea was performed as previously described[31]. Pupae were dissected in cold PBS and trachea were fixed in 4% formaldehyde. After several washes, the samples were permeabilized with 1% TritonX-100, blocked in 10% goat serum, and then incubated with primary antibodies followed by secondary antibodies conjugated to Alexa Fluor 488, 555, or 647. Samples were mounted in Vectashield medium (with DAPI). Images were captured with an LSM Zeiss 900 inverted confocal laser scanning microscope.

Primary antibodies: α-β-galactosidase (mouse, 1:100, Developmental Studies Hybridoma Bank, 1G9), α-Cut (mouse, 1:100, Developmental Studies Hybridoma Bank, 2B10), α-Ncad (mouse, 1:100, Developmental Studies Hybridoma Bank, DN-EX), α-Peb (mouse, 1:100, Developmental Studies Hybridoma Bank, 1G9), α-Delta (mouse, 1:100, Developmental Studies Hybridoma Bank, C594.9B), α-Serp (rabbit, 1:200, gift from Dr. Mark Krasnow), α-Kni (Guinea pig, 1:400), and α-Sal (rabbit, 1:200). Secondary antibodies: α-mouse Alexa Fluor®488 (goat, 1:200, Jackson ImmunoResearch, 115-545-003), α-rabbit Alexa Fluor®488 (goat, 1:200, Jackson ImmunoResearch, 111-545-003), α-mouse Cyanine Cy™3 (goat, 1:200, Jackson ImmunoResearch, 115-165-003), α-rabbit Cyanine Cy™3 (goat, 1:200, Jackson ImmunoResearch, 111-165-003) and α-mouse Alexa Fluor®647 (goat, 1:200, Jackson ImmunoResearch, 115-605-003).

### Live imaging of *Drosophila* trachea
White pupae of *Drosophila* (0 h APF) were briefly washed with PBS and mounted in halocarbon oil 700 (Sigma). Pupae were mixed well with oil, and positioned with forceps so that a single dorsal trunk of the trachea was up for optimal imaging of the Tr4 and Tr5 metameres. Then, pupae were immobilized by a 22×30 mm No.1.5 high precision coverslip spaced by vacuum grease. The time-lapse images were captured by an LSM Zeiss 900 inverted confocal laser scanner microscope with 405 nm, 488 nm, 561 nm, or 640 nm wavelength lasers.

### EdU cell proliferation assay
Pupae were dissected in cold PBS and the fat body and gut were carefully removed. The samples were incubated in 1X EdU solution for 30 min at room temperature and then fixed with 4% formaldehyde in PBS at room temperature for 30 min. The samples were washed with PBS for three times and permeabilized in PBS containing 1% Triton-X-

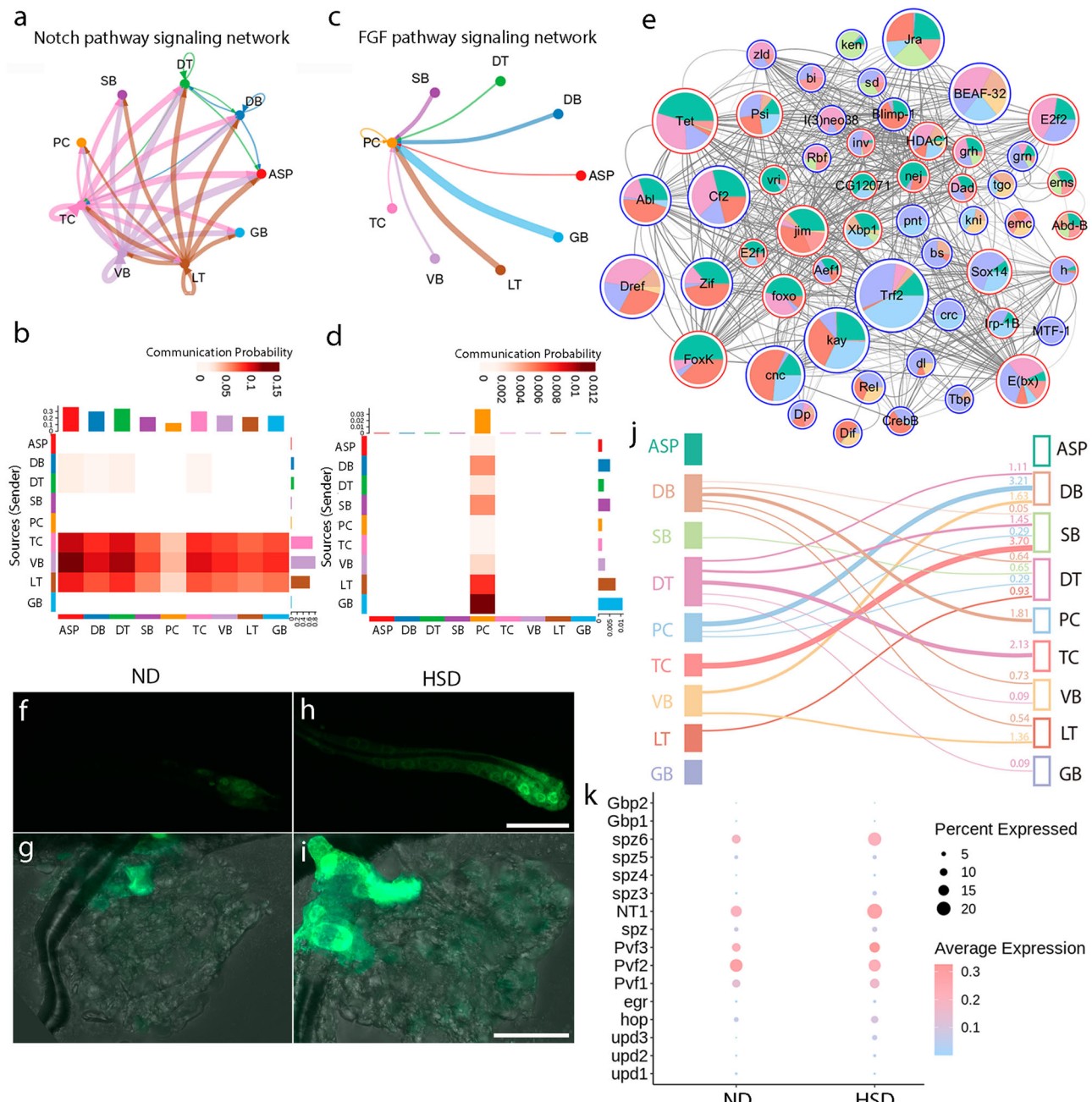

**Fig. 8 | Intercellular communication in trachea cell populations. a, b** Directional Notch signaling between different cell populations. Circle plot (**a**) and heatmap (**b**) showing Notch signaling among individual clusters. **c, d** FGF signaling in tracheal cells. Circle plot (**c**) and heatmap (**d**) depicting the transport of FGF in tracheal cells. **e** Cytoscape diagram depicting the activity of regulons in each cluster. **f–i** Expression of Su(H)-NRE-GFP reporter in DB (**f, h**) and PC (**g, i**) domain. NRE-GFP represents Notch activity under ND (**f, g**) or HSD (**h, i**) condition. Each experiment was repeated independently with similar results for three times. **j** Projection of scRNA-seq dataset from HSD group to ND group. **k** Dotplot representing expression profiles of signaling ligands in the presence or absence of HSD. DT dorsal trunk, DB dorsal branch, TC transverse connective, ASP air sac primordium, VB visceral branch, SB spiracular branch, PC progenitor cells, LT lateral trunk, GB ganglionic branches, ND normal diet, HSD high-sugar diet. Scar bars: 30 μm (**f–i**).

100 for 1 h. Subsequently, the samples were incubated in 5% goat serum in PBS and then treated with Click-iT® reaction cocktail at room temperature for 30 min. After three times of wash, the samples were mounted in Vectashield medium for confocal imaging. Statistics analysis was performed using GraphPad Prism Version 9.5.1. Statistical significance was evaluated with an unpaired two-tailed *t*-test.

### Single-cell RNA-sequencing (scRNA-seq)

The $w^{1118}$ flies were used to harvest tracheal cells. Briefly, 100 tracheae of 0 h APF pupae were dissected in cold Grace medium supplemented with 2.5% fly extract, 1 mM PMSF, and EDTA-free protease inhibitor cocktail (Roche) within 1 h and digested in 1 mg/ml Elastase (Sigma, #E0258) solution at 25 °C for 1 h. Dissociated cells were pelleted at 400 x *g* for 20 min, resuspended in PBS with 0.2% BSA, filtered with 70 mm filters (BD Falcon), and sorted using a FACS Aria III sorter (BD Biosciences). TC20 automated cell counter (BIO-RAD) was used to assess cell number and live/dead cell ratio. About 8000 live cells (ratio of live cells in the suspension >90%) were obtained for 10X genomics library preparation following the manufacturer's manual. Briefly, single cells and GemCode Gel Beads were encapsulated into droplets by using

the GemCode Single-Cell Platform, and within-droplet cell lysis and barcoded reverse transcription reaction were carried out, which generated ~100 ng of cDNA after 12 cycles of reverse transcription. Following standard library preparation, next-generation deep sequencing was then carried out on the Illumina X10 system (Novogene).

### Data pre-processing

Reads in fastq format were aligned to the reference genome *Drosophila melanogaster*.BDGP6.22 by cellranger-6.1.2 command cellranger count respectively. cellranger aggr was then utilized to perform batch effect correction and dataset aggregation. Raw 10X read counts matrices were loaded into 'R' 4.0.2 via Read10X() function of R package 'Seurat' v.4 and then converted to 'Seurat objects'[77]. A filter of nFeature_RNA > 300 & nFeature_RNA < 4000 & nCount_RNA > 1000 & nCount_RNA < 20000 & percent.mt <10 was applied to all of three datasets to remove low-quality cells. SCTransform was used normalize and scale the raw UMI counts to generate the gene expression matrix for each cell. Essentially, a regularized negative binomial model was applied to fit gene expression with the whole scRNA-seq dataset, which minimizes the batch effect caused by differential sequencing depths of individual cells. The parameter 'vars.to.regress' was set to 'percent.mt' to avoid interference from mitochondrial genes.

### Seurat standard workflow

The Seurat standard workflow was applied to the filtered and normalized scRNA-seq datasets. The RunPCA() function was performed with default parameters to obtain expression matrices in a 50-dimension space, then a shared nearest neighbor (SNN) graph was constructed using the top 20 principal components (PCs) via the FindNeighbors() function. Unsupervised partitioning was generated by the Louvain algorithm via the FindClusters() function. A uniform manifold approximation projection of cell populations was generated using the RunUMAP() function based on top20 PCs to visualize cell populations[78]. Individual cell types were annotated according to marker genes. Non-tracheal cells were excluded in downstream analysis via the subset() function.

### Doublet detection

We utilized DoubletFinder v2.0.3 to identify suspicious doublets from scRNA-seq data[79]. The workflow is previously described[80]. The parameters were optimized by the internal function paramSweep_v3.

### Marker gene identification

The FindAllMarkers() and FindMarkers() functions built in the R package Seurat were utilized to identify markers (differentially expressed genes) for each cluster in a reference dataset with unsupervised clusters. Differential expression analysis was performed on genes with more than a 10% fraction difference by the Wilcoxon rank sum test. Genes whose expression level exceeded 0.5 log2 fold enrichment (1.4-fold change) were considered marker genes and were used to generate gene sets in the module scoring function and similarity computation. DESeq2 test was used to compare control and HSD groups. log2 fold enrichment >0.3 and *p*-value ≤ 0.01 was considered as significance.

$$\log 2\text{FoldChange} = \log_2 \frac{\text{average expression in group1}}{\text{average expression in group2}}$$

### Pseudotime analysis

Pseudotime analysis was performed using R package monocle3. The Seurat object was imported into the monocle3 standard workflow. The raw count matrices were processed with the default parameters with the preprocess_cds() and reduce_dimension() functions. Then a graph over all stromal cells was constructed by reversed graph embedding to learn a trajectory via the learn_graph() function. Subsequently,

order_cells() was called to order cells according to pseudotime. The pseudotime value was interpreted as the transcriptomic similarity between the target cell and PC.

### CellChat

Intercellular communication was inferred by the expression of ligands and receptors, using CellChat package[41]. When computing the communication strength between interacting cell groups, a communication filter was set with parameters min.cells = 10. Population communication with less than 10 cells was excluded.

### Gene ontology

The enrichGO() function in R package clusterProfiler recognized a list of marker genes from a given cell population and produced an enrich.go object[81]. The annotation was performed using org.Dm.eg.db 3.11.4. The enrich.go objects were visualized by dotplot() and barplot() in R package enrichplot.

### SCENIC

The SCENIC analysis was performed with standard workflow[82]. GENIE3 was utilized to infer the gene co-expression network. RcisTarget was called for the enrichment of transcription factor binding motifs. AUCell was used to compute the transcription factors activity area under the curve (AUC) of each cell in order to identify cells with active gene sets modulated by specific transcription factors (TFs). TFs activity matrix among different clusters was visualized by Complex-Heatmap::Heatmap() function[83].

### RNA sequencing of tracheal progenitors

The L3 larvae, white pupae (0 h APF), or 2-h-APF pupae were dissected in cold PBS and a single cluster of progenitors from Tr5 metamere was subjected to RNA extraction using the RNeasy Micro Kit from Qiagen. The total RNA from each sample was used for sequencing library preparation. Three biological replicates were performed for each genotype or treatment. The SMART-Seq v4 Ultra-low input RNA Kit was used for first-strand and second-strand cDNA synthesis and double-stranded cDNA end repair. Double-stranded cDNAs were purified with the AMPure XP beads from Beckman Coulter, subjected to tagmentation, and ligated to adaptors. Finally, the libraries were generated by PCR enrichment of the adaptor-ligated DNA. The concentration and quality of the constructed sequencing libraries were measured by using the Agilent High Sensitivity DNA Kit and a Bioanalyzer 2100 from Agilent Technologies. The libraries were submitted to the Hiseq4000 sequencer with 150 bp paired-end mode.

RNA-seq data analysis was performed with the standard pipeline. The clean reads were mapped to the *Drosophila* genome sequence using Hisat2 with default parameters. The number of mapped reads was counted by featureCounts. Differential gene expression analysis was performed using the DESeq2 package. Adjusted *p*-value < 0.05 was used as the threshold to identify the differentially expressed genes. Gene ontology and KEGG pathway enrichment analyses for the differentially expressed genes were conducted using the Database for Annotation, Visualization, and Integrated Discovery (DAVID).

### *Gal80^ts* inactivation

The expression of btl-Gal4 was restricted by temperature-sensitive *tub-Gal80^ts*. Larvae expressing *UAS-Notch^DN*, *UAS-mmyRNAi*, *UAS-Tom*, *UAS-sal*, *UAS-kni*, *UAS-fngRNAi*, *UAS-O-fut1RNAi*, *tGPH*, *ex-lacZ*, *dad-GFP* were raised at 18 °C and then shifted to non-permissive temperature of 29 °C for 72 h. White pupae were collected for dissection and imaging.

### Detection of GlcNAcylation

Notch-GFP-expressing trachea from *n* = 150 ND/HSD-fed white pupae were lysed in 220 μl homogenization buffer (125 mM NaCl, 50 mM HEPES, 5% NP-40, 1% TritonX-100, 100 μM TMG, pH 7.9). 210 μl

supernatant per sample was collected, of which 10 µl was reserved as loading control. Then, supernatants were incubated with UDP-GalNAz and Gal-T1 (Y289L) enzyme from the Click-it GlcNAc enzymatic labeling system (Thermo Fisher, #C33368). In the enzymatic reaction, Gal-T1 transfers the azide-modified galactose (GalNAz) from UDP-GalNAz to GlcNAc-modified proteins in the supernatant. Subsequently, the azide-modified products were labeled by biotin using a Click-it glycoprotein detection kit (Thermo Fisher, #C33372). Streptavidin magnetic beads (Thermo Fisher, #88816) were used to capture the biotin-labeled GlcNAcylated proteins. The loading control and biotin-bound GlcNAcylated proteins were separated by SDS/PAGE on 4–15% Bis-Tris gels and followed by immunoblot analysis. Primary antibody: α-GFP (rabbit, 1:2000, Invitrogen, #A-11122). Secondary antibody: HRP-conjugated α-rabbit (goat, 1:5000, Abcam, #ab6721).

### Detection of O-GlcNAc

ND/HSD-fed ($n = 80$) white pupae (w1118) were lysed in 80 µl homogenization buffer (1x PBS, 5% NP-40, 1% TritonX-100) with protease inhibitor (Roche). The supernatant was collected and then subjected to SDS/PAGE on 4–15% Bis-Tris gels followed by immunoblot analysis. Primary antibody: α-O-GlcNAc (CTD110.6, mouse, 1:1000, CST, #9875). Secondary antibody: HRP-conjugated α-mouse (rabbit, 1:5000, ABclonal).

### Immunoprecipitation

Protein extracts were prepared from $n = 150$ ND/HSD-fed white pupae expressing Notch-GFP with 300 µl homogenization buffer (1x PBS, 5% NP-40, 1% TritonX-100) containing protease inhibitor (Roche). 20 µl anti-GFP magnetic beads were added to the lysate. The samples were incubated at 4 °C for 16 h under gentle agitation. The beads were washed for at least three times. Finally, the proteins were eluted with 5 × SDS loading buffer for SDS/PAGE and followed by immunoblot analysis. Primary antibody: α-O-GlcNAc (CTD110.6) (mouse, 1:1000, CST, #9875) and α-GFP (rabbit, 1:2000, Invitrogen, #A-11122). Secondary antibody: HRP-conjugated α-mouse (rabbit, 1:5000, ABclonal) and HRP-conjugated α-rabbit (goat, 1:5000, Abcam, #ab6721).

### Lineage tracing

A *UAS-G-TRACE* stock (BDSC 28280) was crossed to *kni*-Gal4, *cut*-Gal4, or *esg*-Gal4 at 18 °C. The F1 progeny fed with ND/HSD were kept at 18 °C till 2nd instar and then shifted to non-permissive temperature (29 °C) to allow activation of the Gal4 for 2 days. The trachea from white pupae was dissected and fixed in 4% paraformaldehyde. After several washes, the trachea was permeabilized with 1% TritonX-100 and mounted in Vectashield (with DAPI). Images were captured by an LSM Zeiss 900 inverted confocal laser scanning microscope.

### Reporting summary

Further information on research design is available in the Nature Portfolio Reporting Summary linked to this article.

## Data availability

The authors declare that all data supporting the present study, including its supplementary information files, are available within this article. The SMART-Seq data (L3, 0 h APF, and 2 h APF) generated and analyzed in this study have been deposited in the NCBI database under accession number GSE184856. The single-cell RNA-sequencing data and the SMART-Seq data (control and *mmyRNAi*) generated and analyzed in this study have been deposited in the NCBI database under accession number GSE240777. Source data are provided in this paper.

## Code availability

All custom scripts are available on GitHub [https://github.com/Tianfeng-Lu/single-cell-atlas-of-fly-trachea] and Zenodo [https://doi.org/10.5281/zenodo.10672045][84].

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

## Acknowledgements

We thank Drs. Thomas Kornberg, Xiaohang Yang, Markus Affolter, Mark Krasnow, Stefan Luschnig, Kai Yuan, Zhouhua Li, Sarah Bray, and Yiming Zheng for generously providing reagents; Drs Wen Yi and Qiang Zhu for support in glycosylation analysis; Dr. Yingnan Bai for support in bioinformatics; Core Facility of Drosophila Resource and Technology, SIBCB, CAS for injection service; Bloomington Drosophila Stock Center, Kyoto Stock Center, Vienna stock center, Tsinghua Stock Center for fly stocks; Developmental Studies Hybridoma Bank for antibodies; all members of Huang lab and Kornberg lab for discussions and constructive suggestions. This work has been financially supported by NSFC92168101, NSFC32070784, and Thousand Young Talent Program to H.H., NSFC32000574 to H.G.W., and NSFC32300699, LQ24C120001 and Postdoctoral Fellowship Foundation 2023M733098 to Y.L.

## Author contributions

Conceptualization, H.H., Y.L.; Bioinformatics, formal analysis, and software, T.F.L., Q.Y.Z., H.G.W; Investigation and interpretation, Y.L., H.H., T.F.L., P.Z.D., J.C., Q.Z., Y.Y.W., T.H.X.; Writing, H.H., Y.L., T.F.L., Q.Y.Z., H.G.W.; Supervision, H.H., Q.Y.Z., H.G.W.

## Competing interests

The authors declare no competing interests.
