## [Peer Review File · Nature Communications]

A single-cell atlas of *Drosophila trachea* reveals glycosylation-mediated Notch signaling in cell fate specificationReviewers' Comments:

Reviewer #1:

Remarks to the Author:

Li and colleagues carry out a single cell seq analysis of tracheal cells dissected out of early pupa, identifying 9 major tracheal cell subtypes based on cluster analysis. The 9 major subtypes corresponded to different branches of the tracheal system. Authors found that prior gene expression analysis was consistent with subtype transcriptional profiles – for instance, the progenitor cell cluster showed high expression of blistered. Three additional clusters are attributed to contaminating tissues (fat body, muscle and neuroendocrine).

Authors identify multipotent tracheoblasts as belonging to dorsal branch (DB) and progenitor cell (PC) clusters.

Overall, I find the manuscript to offer worthwhile additions and believe it will be of interest to the community, but would like the authors to respond to the following questions/concerns:

> there are proliferative DT cells in TR2 that contribute to pupal/adult trachea, where do those cells cluster? are they multipotent?

> So far as I can determine, Gal4/UAS experiments would have caused k/d or overexpression from embryonic stages through to the early pupal stage analyzed. How can early and late effects be disentangled.

> Can authors clarify which RNAi phenotypes are new and which have been previously published (or if there previously published loss of function data).

> Can authors elaborate on the "muscle related" subclusters of progenitor cells. What are these?

> For glycosylation mutants, can authors put findings into the context of prior work -- Tian E, Ten Hagen KG. A UDP-GalNAc:polypeptide N-acetylgalactosaminyltransferase is required for epithelial tube formation. J Biol Chem. 2007 Jan 5;282(1):606-14. doi: 10.1074/jbc.M606268200. Epub 2006 Nov 10. PMID: 17098739.

> Can authors provide evidence that glycosylation mutants do not interfere with other signaling pathways -- Tian and Ten Hagen report suggests that there might be vesicle trafficking defects that might disrupt most signaling pathways...

> kni-GAL4 does not seem to be described in the cited reference, can author's clarify? Perhaps I missed it.

> Can authors clarify advances vs. Rao et al

Reviewer #2:

Remarks to the Author:

In this study the authors take multiple genomic, bioinformatic, and genetic approaches to investigate the composition of Drosophila trachea and the programs that guide their differentiation. Their approach is extremely thorough and revealed an important and previously unappreciated effect of diet on Notch-mediated regulation of cell fate in this tissue. This is an interesting finding that provides new insight into the connections between metabolism, signaling, and cell differentiation. In addition, the single cell sequencing data is a useful resource for the community on its own and the many different forms of bioinformatic analysis performed not only make excellent use of these data, but also provide

an example that can be followed for analysis of other single cell datasets. Overall, I believe that this study produced significant advances that would be of broad interest. However, there are several concerns that should be addressed before publication

1. The bioinformatic analysis is made available through a Github repository but the code is very poorly organized and annotated. In its current state, it is almost entirely unusable by anyone other than the original authors, which is unacceptable, especially for an open source journal such as Nature Communications. Therefore, the code should be substantially rewritten and reorganized so that it complies with the basic standards of reproducibility in computational research (e.g. see Peng, 2011, DOI: 10.1126/science.1213847). Some examples of changes that need to be made are that a README file should be included that describes the code in detail; the code should be reorganized so that it uses a consistent file structure for the input files and the code executable when the repository is cloned; and annotations in the code or the README file should make it clear which code was used to generate each of the panels in all of the figures.
2. In Figure 2, the immunostainings show strong, near-uniform staining of the selected marker in the indicated branch and complete absence of staining on other branches, but the patterns shown on the UMAP plots in this figure suggest that the indicated clusters are more heterogeneous for these markers. This could be, for example, because the clustering and annotation needs further refinement, or because there are doublets in the dataset, or perhaps these markers really are heterogeneous in these cell populations. This heterogeneity is noted by the authors and explored in Fig. 3 for the PC, but not for the other clusters. In addition, the G-TRACE experiments in Fig. 7 suggest that *kni* expressing cells do not contribute to the DT in normal feeding conditions, but Fig. 2b shows some *kni*⁺ cells in the DT region. The authors should address this heterogeneity in the other clusters with additional data, data analysis, and/or explanation in the methods. Showing the expression of these key markers across the different clusters in upset plots, as in Supplementary Fig. 2, and/or violin plots would also be helpful.
3. Also, it is useful to see the comparison between the branch that is predicted to express the indicated marker and other branches that are predicted to not express this marker in Fig. 2e, f, j, and l. The same should be done for Fig. 2d and k.
4. The differences in the level of Delta signaling between Fig. 4a, c, and e are not convincing. These differences should be quantified and better characterized.
5. The Western blot in Fig. S6 is not very convincing. These bands should be quantified and, ideally, a cleaner blot should be shown.
6. Characterizing the observation that ectopic expression of Tom in DB endowed DB cells with the “molecular signature of DT” is an overstatement since only a single marker (Delta), not an entire molecular signature, was assessed in this experiment.
7. Likewise, it is an overstatement to imply that the results of Cellchat can be used to conclude that “progenitors received inputs of Notch signaling from multiple branches”. This is a prediction of the algorithm that would need to be tested experimentally before a firm conclusion can be reached.
8. I do not understand why the dashed line in Fig. 5m is drawn where it is. It seems to be bisecting a branch and to include multiple distinct regions. Likewise, it would be helpful to show a DAPI stain or other marker so the reader can see the shape of the tissue that is bound by the dashed line in Fig. 5n (and Fig. 5m).

Reviewer #3:

Remarks to the Author:

Authors conducted single-cell RNA sequencing of *Drosophila* tracheal cells during development and revealed heterogeneous cell populations. Furthermore, the authors demonstrated the involvement of Notch signaling in the differentiation processes. However, the data analysis related to glycosylation appears somewhat misleading or incomplete. Particularly, the data for *o-fut1*, *fng*, and *mmy* RNAi require further analysis. Additionally, the effect of a sucrose-enriched high-sugar diet remains unclear, as the authors examined *fng* (which modifies O-fucose) and *kud* (which modifies N-glycans) in RNA-seq, whereas O-GlcNAc expression was analyzed using immunoblotting. Below, I have summarized my major and minor concerns primarily related to the glycosylation analysis.

1. Figure 6a: 'Protein glycosylation', 'macromolecule glycosylation', and 'glycosylation' would essentially indicate the same gene sets. Redundant information should be removed.
2. Figure 6c: The signal corresponding to *O-fut1* in GB looks like missing. Does this mean no expression of *O-fut1* in GB cells? It should be noted that *O-fut1* is indispensable for Notch signaling.
3. Figure 6d contains incorrect information that severely affects the interpretation of data of this study. OGT (encoded by *sxc*) catalyzes protein O-GlcNAcylation. *O-fut1* catalyzes O-fucosylation of EGF domains, and *fng* catalyzes GlcNAc modification of O-fucose. Thus, the glycosylation pathways shown in the figure are incorrect. It is noted that EOGT catalyzes O-GlcNAc modification of EGF domains. It should also be noted that *kud* (N-glycosylation) is not described in this figure.
4. Figures 6f-m: It is unclear whether the authors intend to highlight the cell numbers of GFP-positive cells or the intensity of GFP signals in this figure. I suggest the authors directly measure the expression of Notch downstream genes, as analyzed in Figure 5g-j. Moreover, it appears that *mmy* RNAi affects the cell size of GFP-positive cells. Given that UDP-GlcNAc synthesis mediated by UAP1 (*mmy*) affects various GlcNAc modifications, including N-glycans, O-glycans, O-GlcNAc, and others, readers might wonder whether the lack of GlcNAc impairs cell proliferation or viability.
5. Figure 6o: The reviewer did not fully comprehend the experimental design related to this figure. In the methods section, the authors mentioned that they aimed to detect O-GlcNAc on Notch, which is not clearly depicted in the figure (i.e., it should be O-GlcNAc instead of GlcNAcylation). Furthermore, it is unclear how GlcNAcylation (O-GlcNAc) is detected by immunoblotting shown in the top panels. Since GalNAz could be converted to other sugar moieties besides O-GlcNAc, independent detection of O-GlcNAc with an O-GlcNAc antibody (e.g., CTD110.6) would be necessary and is experimentally feasible. If O-GlcNAc increases upon a sucrose-enriched high-sugar diet (HSD), how does it affect Notch activity?
6. O-GlcNAc blot in extended data fig. 6 shows no apparent difference between control and HSD.

REVIEWER COMMENTS

Reviewer #1 (Remarks to the Author):

Li and colleagues carry out a single cell seq analysis of tracheal cells dissected out of early pupa, identifying 9 major tracheal cell subtypes based on cluster analysis. The 9 major subtypes corresponded to different branches of the tracheal system. Authors found that prior gene expression analysis was consistent with subtype transcriptional profiles – for instance, the progenitor cell cluster showed high expression of blistered. Three additional clusters are attributed to contaminating tissues (fat body, muscle and neuroendocrine).

Authors identify multipotent tracheoblasts as belonging to dorsal branch (DB) and progenitor cell (PC) clusters.

Overall, I find the manuscript to offer worthwhile additions and believe it will be of interest to the community, but would like the authors to respond to the following questions/concerns:

> there are proliferative DT cells in TR2 that contribute to pupal/adult trachea, where do those cells cluster? are they multipotent?

Authors' response: The cells that constitute the Tr2 branches reinitiate cell division during the L3 stage to increase the Tr2 population and are considered as multipotent cells¹. We annotated DT cells into three subclusters using FindSubCluster (Extended Data Figure 6), and found that a subpopulation of DT cells, DT2, was remarkably different from other two subpopulations (Extended Data Figure 6a). In our scRNA-seq data, there are developmental trajectories from DT2 to SB or DT0 populations, suggesting that DT2 population might be multipotent (Extended Data Figure 6b,c). It should be mentioned that previous publications reported proliferating cells in Tr2 dorsal trunk during L3 stage, whereas our single-cell preparation was generated from white pupae.

> So far as I can determine, Gal4/UAS experiments would have caused k/d or overexpression from embryonic stages through to the early pupal stage analyzed. How can early and late effects be disentangled.

Authors' response: We agree with you that the expression of these drivers in embryonic stages should be excluded. We performed additional experiments with *tub-Gal80^{ts}* and expression of UAS construct is only in larval stage, e.g. Figure 4, Figure 6, Extended Data Figure 8, and Extended Data Figure 9. All the transformation experiments are conducted with *tub-Gal80^{ts}*.

> Can authors clarify which RNAi phenotypes are new and which have been previously published (or if there previously published loss of function data).

Authors' response: The tracheal progenitors have not been extensively explored. Hence, the regulators of progenitors did not appear in any previous publications and represent new findings in this study. Although it is reported that loss of function of *exp*², *kni*³ or *bs*⁴ causes general defects in tracheal development, a detailed phenotypical analysis has not been performed in progenitor cells or DB cells. The additional molecular features such as *mirr*, *bru2* and *wun* are first characterized by present study.

> Can authors elaborate on the "muscle related" subclusters of progenitor cells. What are these?

Authors' response: The muscle-related subpopulation of progenitors appeared to express higher level of *mysospheroid* (*mys*), a marker gene functionally related to muscle, and be present at the leading front of migratory progenitors, as shown in Extended Data Figure 4h-i'. We also detected the expression of *Osi15*, the marker gene of muscle-related subclusters in the majority of PCs. Their expressions were pronounced in the PCs that attached to DT and those at the frontal edge (see Extended Data Figure 4j-k'). Furthermore, these PC populations exhibited functional requirement of muscle. Depletion of muscles in L3 stage by expressing pro-apoptotic cell death genes, *hid* and *reaper* (*rpr*), caused reduction of proliferation of tracheal progenitors, but did not alter cut-expressing PCs (Extended Data Figure 4f,g).

> For glycosylation mutants, can authors put findings into the context of prior work -- Tian E, Ten Hagen KG. A UDP-GalNAc:polypeptide N-acetylgalactosaminyltransferase is required for epithelial tube formation. J Biol Chem. 2007 Jan 5;282(1):606-14. doi: 10.1074/jbc.M606268200. Epub 2006 Nov 10. PMID: 17098739.

Authors' response: We thank you for pointing out galactose (Gal) modification that has been reported on mammalian Notch. This modification contributes to Notch signaling in a mechanism that may be other than modulation of Notch-ligand binding⁵. We detected considerable expression of an *N*-acetylgalactosamine (GalNAc) transferase, *Pgant35A*, in different clusters of tracheal cells (Fig. 6c,d). Furthermore, the expression of *Pgant35A* is enhanced by high sugar diet (HSD) (Fig. 6r).

> Can authors provide evidence that glycosylation mutants do not interfere with other signaling pathways -- Tian and Ten Hagen report suggests that there might be vesicle trafficking defects that might disrupt most signaling pathways...

Authors' response: With regard to your suggestion, we performed RNA-seq analysis of tracheal progenitors from control and *mmvRNAi* flies, and identified differentially expressed genes (DEGs). In addition to Notch signaling pathway characterized and substantiated by present study, Gene Ontology (GO) analysis also revealed signaling pathways such as Hippo, TGF- β , and Insulin/FOXO (Extended Data Fig. 8a). To determine if these signaling pathways are altered upon the depletion of glycosylation by expressing *mmvRNAi*, we examined a variety of reporters. The expressions of ex-

lacZ for Hippo signaling, tGPH for insulin signaling, and *dad*-GFP for Dpp/TGF- β signaling pathway were unaltered upon the expression of *mmyRNAi*, suggesting that reduction of GlcNAcylation did not perturb Hippo, insulin or Dpp/TGF- β signaling (Extended Data Fig. 8b-j).

> *kni*-GAL4 does not seem to be described in the cited reference, can author's clarify? Perhaps I missed it.

Authors' response: we corrected it and cited the original reference - The fatty acyl-CoA reductase Waterproof mediates airway clearance in *Drosophila*.

> Can authors clarify advances vs. Rao et al

Authors' response: Rao et al. reported developmental compartments in the larval tracheal tubes. Specifically, Rao and our colleagues in Kornberg lab described several molecular signatures of tracheal compartments of Tr2 metamere and alarmed community for the heterogeneity of tracheal epithelium. For instance, Spalt, Delta in Dorsal Trunk (DT), Knirps in Dorsal Branch (DB), etc. The domains of gene expression respect lineage restriction borders. The previous discovery, although unable to resolve the entire spectrum of molecular signatures with conventional methods, brought up the importance and interest of tracheal cell atlas.

Our present investigation has been designed to elaborate the transcriptomic atlas of fly airway. Compared with previous studies, a major advance in the present study is the illustration of transcriptomes and regulatory networks of all major cell types as well as subpopulations during *Drosophila* tracheal morphogenesis, which provides an opportunity to analyze the coordination of differentiation and proliferation during morphogenesis of the tracheoblasts. In combination with bulk RNA-seq at different stages, the atlas also delineates the key regulators that control tracheal cell identities, provides mechanistic insights into high sugar-induced glycosylation, and suggests that a glycosylation-mediated Notch signaling is responsible for cell fate determination during tracheal development. Furthermore, the developmental trajectory, cell fate determination and functional diversification of tracheal progenitors investigated by present study are irrelevant to Rao et al. and unrecognized by other reports. We believe these findings provide insightful messages in glycobiology and stem cells.

Reviewer #2 (Remarks to the Author):

In this study the authors take multiple genomic, bioinformatic, and genetic approaches to investigate the composition of *Drosophila* trachea and the programs that guide their differentiation. Their approach is extremely thorough and revealed an important and previously unappreciated effect of diet on Notch-mediated regulation of cell fate in this tissue. This is an interesting finding that provides new insight into the connections between metabolism, signaling, and cell differentiation. In addition, the single cell sequencing data is a useful resource for the community on its own and the many

different forms of bioinformatic analysis performed not only make excellent use of these data, but also provide an example that can be followed for analysis of other single cell datasets. Overall, I believe that this study produced significant advances that would be of broad interest. However, there are several concerns that should be addressed before publication

1. The bioinformatic analysis is made available through a Github repository but the code is very poorly organized and annotated. In its current state, it is almost entirely unusable by anyone other than the original authors, which is unacceptable, especially for an open source journal such as Nature Communications. Therefore, the code should be substantially rewritten and reorganized so that it complies with the basic standards of reproducibility in computational research (e.g. see Peng, 2011, DOI: 10.1126/science.1213847). Some examples of changes that need to be made are that a README file should be included that describes the code in detail; the code should be reorganized so that it uses a consistent file structure for the input files and the code executable when the repository is cloned; and annotations in the code or the README file should make it clear which code was used to generate each of the panels in all of the figures.

Authors' response: In accordance with your kind suggestion, we have updated our GitHub repository to ensure that all our code complies with the standards of reproducibility. Despite using only well-established pipelines and functions without developing any novel computational approaches in this manuscript, we have thoroughly revised the code throughout the entire R script. Additionally, we have deposited an interactive R Jupyter notebook and a download script for both the raw and processed data to ensure proper reproducibility of all the results in this manuscript with a simple repository clone on GitHub.

Furthermore, we have revised the README file, providing detailed explanations of the function and input/output for each script block of code. We have also annotated the code for each panel in all the figures.

2. In Figure 2, the immunostainings show strong, near-uniform staining of the selected marker in the indicated branch and complete absence of staining on other branches, but the patterns shown on the UMAP plots in this figure suggest that the indicated clusters are more heterogeneous for these markers. This could be, for example, because the clustering and annotation needs further refinement, or because there are doublets in the dataset, or perhaps these markers really are heterogeneous in these cell populations. This heterogeneity is noted by the authors and explored in Fig. 3 for the PC, but not for the other clusters. In addition, the G-TRACE experiments in Fig. 7 suggest that *kni* expressing cells do not contribute to the DT in normal feeding conditions, but Fig. 2b shows some *kni*⁺ cells in the DT region. The authors should address this heterogeneity in the other clusters with additional data, data analysis, and/or explanation in the methods. Showing the expression of these key markers across the different clusters in upset plots, as in Supplementary Fig. 2, and/or violin

plots would also be helpful.

Authors' response: Regarding your kind suggestion, we extended our subpopulation identification to other clusters and further revealed heterogeneity in these tracheal populations (Extended Data Figure 5). For instance, DT cells were annotated into three subclusters, in which DT2 might be multipotent (Extended Data Figure 6). The upset plots in revised Extended Data Figure 2 show the expressions of key markers across the different clusters, and suggest the heterogeneity within each cluster. For ~10% *Kni*⁺ cells in DT in Extended Data Figure 2b as you kindly pointed out, we agree with you that inevitable doublets may disturb homogeneity of markers, although the number of doublets in our samples was relatively small according to DoubletFinder.

3. Also, it is useful to see the comparison between the branch that is predicted to express the indicated marker and other branches that are predicted to not express this marker in Fig. 2e, f, j, and l. The same should be done for Fig. 2d and k.

More upset and violin plots like

Authors' response: Regarding your kind suggestion, we added the comparison between the branch that is predicted to express the indicated marker and other branches that are predicted to not express this marker in Fig. 2. The upset plots in revised Extended Data Figure 2 show the expression of indicated markers across the different clusters. *kni* is predominantly expressed in DB cells, as shown in Fig. 2e. The upset plots show that *kni* is expressed in majority of DB cells and has a broad expression in PCs. Its expression is below 10% in other branches. Fig. 2f,j show the expression of *ct* in TC and SB and the expression of *wg* in SB. In agreement with these immunostaining results, *ct* is predominantly expressed in TC and SB branches, as shown in Extended Data Figure 2d,e. The expression of *wg* is only present in SB but not other clusters in upset plots (Extended Data Figure 2). Fig. 2l shows that *bs* is a marker gene for PCs and its expression is undetectable in DT, TC or VB cells. According to the prediction, the expression level of *bs* is below 5% in other branches (Extended Data Figure 2). The upset plots in Extended Data Figure 2 predict that *spalt* (*sal*) is highly expressed in DT cells, and is also expressed in VB, but is barely detected in DB or PC cells, which is consistent with the expression pattern of *sal-Gal4* in Fig. 2d. The expression of *vn* is present in VB cells, but is absent in SB, PC or TC (Extended Data Figure 2d-f), which is in agreement with the expression of *vn-nlslacZ* in Fig. 2k.

4. The differences in the level of Delta signaling between Fig. 4a, c, and e are not convincing. These differences should be quantified and better characterized.

Authors' response: According to your suggestion, we repeated the experiments for several times and qualified the staining results with a considerable number of replicates. The quantification results are shown in Fig. 4m,n.

5. The Western blot in Fig. S6 is not very convincing. These bands should be

quantified and, ideally, a cleaner blot should be shown.

Authors' response: We performed additional experiments and quantified bands as you kindly suggested, as shown in revised Extended Data Figure 11.

6. Characterizing the observation that ectopic expression of Tom in DB endowed DB cells with the “molecular signature of DT” is an overstatement since only a single marker (Delta), not an entire molecular signature, was assessed in this experiment.

Authors' response: We corrected the statement and thank you for kindly pointing it out.

7. Likewise, it is an overstatement to imply that the results of Cellchat can be used to conclude that “progenitors received inputs of Notch signaling from multiple branches”. This is a prediction of the algorithm that would need to be tested experimentally before a firm conclusion can be reached.

Authors' response: We agree with you that additional experiments are required to confirm the results of Cellchat. Hence, we reduced Notch ligand, Delta, in TC and SB using *ct-Gal4*, or in DT using *sal-Gal4*. The results showed that reduction of Notch ligands in TC, SB or DT by expressing *DIRNAi* impaired Notch signaling in the progenitors, as assayed by Peb staining, which further validated that TC, SB and DB are Notch signaling sending cells for progenitors (see Extended Data Fig. 7). We also modified statement as you kindly suggested.

8. I do not understand why the dashed line in Fig. 5m is drawn where it is. It seems to be bisecting a branch and to include multiple distinct regions. Likewise, it would be helpful to show a DAPI stain or other marker so the reader can see the shape of the tissue that is bound by the dashed line in Fig. 5n (and Fig. 5m).

Authors' response: According to your kind suggestion, we outlined region of progenitors based on DAPI staining. In the revised Figure 5, we included merge channels that contain both bs-GFP and DAPI.

Reviewer #3 (Remarks to the Author):

Authors conducted single-cell RNA sequencing of *Drosophila* tracheal cells during development and revealed heterogeneous cell populations. Furthermore, the authors demonstrated the involvement of Notch signaling in the differentiation processes. However, the data analysis related to glycosylation appears somewhat misleading or incomplete. Particularly, the data for *o-fut1*, *fng*, and *mmy* RNAi require further analysis. Additionally, the effect of a sucrose-enriched high-sugar diet remains unclear, as the authors examined *fng* (which modifies O-fucose) and *kud* (which modifies N-glycans) in RNA-seq, whereas O-GlcNAc expression was analyzed using immunoblotting. Below, I have summarized my major and minor concerns primarily related to the glycosylation analysis.

1. Figure 6a: 'Protein glycosylation', 'macromolecule glycosylation', and 'glycosylation' would essentially indicate the same gene sets. Redundant information should be removed.

Authors' response: We removed redundant information, according to your suggestion.

2. Figure 6c: The signal corresponding to O-fut1 in GB looks like missing. Does this mean no expression of O-fut1 in GB cells? It should be noted that O-fut1 is indispensable for Notch signaling.

Authors' response: There are ~10% of GB cells expressing O-fut1. Unfortunately, they are not clearly shown in our previous representation. In revised figures, we provide enlarged plot for O-fut1 (see revised Figure 6d).

3. Figure 6d contains incorrect information that severely affects the interpretation of data of this study. OGT (encoded by *sxc*) catalyzes protein O-GlcNAcylation. O-fut1 catalyzes O-fucosylation of EGF domains, and *fng* catalyzes GlcNAc modification of O-fucose. Thus, the glycosylation pathways shown in the figure are incorrect. It is note that EOGT catalyzes O-GlcNAc modification of EGF domains. It should also be noted that *kud* (N-glycosylation) is not described in this figure.

Authors' response: We corrected the schematics and thank you for kindly pointing it out. In the revised Figure 6e, we included the information of EOGT and *kud*.

4. Figures 6f-m: It is unclear whether the authors intend to highlight the cell numbers of GFP-positive cells or the intensity of GFP signals in this figure. I suggest the authors directly measure the expression of Notch downstream genes, as analyzed in Figure 5g-j. Moreover, it appears that *mmy* RNAi affects the cell size of GFP-positive cells. Given that UDP-GlcNAc synthesis mediated by UAP1 (*mmy*) affects various GlcNAc modifications, including N-glycans, O-glycans, O-GlcNAc, and others, readers might wonder whether the lack of GlcNAc impairs cell proliferation or viability.

Authors' response: To directly measure the expression of Notch target genes as you kindly suggested, we examined the expression of Notch targets in *mmy* RNAi flies with *tub-Gal80^{ts}* to achieve inactivation of *mmy* in L3 stage. The results showed that the expression of Notch downstream genes, such as *E(spl)m β -HLH*, *E(spl)m2-BFM* and *E(spl)m7-HLH*, were decreased upon the expression of *mmy* RNAi (see Fig. 6g-i). We also provided quantification for GFP signal of Notch reporter, as shown in Extended Data Figure 9a-d. To determine if reduction of UAP1 (*mmy*) impairs cell proliferation or viability, we performed EdU cell proliferation assay and assessed cell viability by immunostaining with cleaved Caspase-3 antibodies. Extended Data Figure 9e-g show that neither cell viability nor proliferation was altered by the expression of *mmy* RNAi.

5. Figure 6o: The reviewer did not fully comprehend the experimental design related to this figure. In the methods section, the authors mentioned that they aimed to detect

O-GlcNAc on Notch, which is not clearly depicted in the figure (i.e., it should be O-GlcNAc instead of GlcNAcylation). Furthermore, it is unclear how GlcNAcylation (O-GlcNAc) is detected by immunoblotting shown in the top panels. Since GalNAz could be converted to other sugar moieties besides O-GlcNAc, independent detection of O-GlcNAc with an O-GlcNAc antibody (e.g., CTD110.6) would be necessary and is experimentally feasible. If O-GlcNAc increases upon a sucrose-enriched high-sugar diet (HSD), how does it affect Notch activity?

Authors' response: We apologize for inadequate description of Click chemistry experiments. The protein extracts of trachea from Notch:GFP pupae were incubated with UDP-GalNAz and Gal-T1 (Y289L) enzyme. GlcNAcylation (GlcNAc) is converted and recognized by Click-it GlcNAc enzymatic labeling system. During the reactions, Gal-T1(Y289L) enzyme transfers the azide-modified galactose (GalNAz) from UDP-GalNAz to GlcNAc-modified proteins such as Notch. Then, enzymatically azide-modified proteins were further labeled by biotin in an azide/alkyne reaction. The biotin-labeled proteins were captured by streptavidin magnetic beads and the abundance of Notch:GFP in the elution were measured by immunoblotting with α -GFP antibodies. We have modified the method part accordingly with detailed procedure. We have also made efforts to revise labels in the figure to avoid misconception.

Furthermore, we performed the suggested experiments to detect O-GlcNAc on Notch using an O-GlcNAc antibody. The Notch proteins were enriched by GFP tag. The immunoblotting showed that the abundance of O-GlcNAc-bearing Notch is higher in HSD group than that of ND group (see Extended Data Figure 11b).

We showed that HSD enhanced expression of *fng*, *mmy* which contribute to the glycosylation of Notch receptor. Glycosylation of Notch receptors mediated by Fng is essential for its activity^{6,7}. Mechanistically, it is reported that glycosylation of Notch receptors enhances their binding to Delta ligand^{5,6,8}.

6. O-GlcNAc blot in extended data fig. 6 shows no apparent difference between control and HSD.

Authors' response: We repeated this experiment for several times and add quantifications, as shown in Extended Data Fig. 11a.

- 1 Guha, A., Lin, L. & Kornberg, T. B. Organ renewal and cell divisions by differentiated cells in *Drosophila*. *Proc Natl Acad Sci U S A* **105**, 10832-10836, doi:10.1073/pnas.0805111105 (2008).
- 2 Iordanou, E. *et al.* The novel Smad protein Expansion regulates the receptor tyrosine kinase pathway to control *Drosophila* tracheal tube size. *Dev Biol* **393**, 93-108, doi:10.1016/j.ydbio.2014.06.016 (2014).
- 3 Chen, C. K. *et al.* The transcription factors KNIRPS and KNIRPS RELATED control cell migration and branch morphogenesis during *Drosophila* tracheal development. *Development* **125**, 4959-4968 (1998).

- 4 Gervais, L. & Casanova, J. The Drosophila homologue of SRF acts as a boosting mechanism to sustain FGF-induced terminal branching in the tracheal system. *Development* **138**, 1269-1274, doi:10.1242/dev.059188 (2011).
- 5 Xu, A. *et al.* In vitro reconstitution of the modulation of Drosophila Notch-ligand binding by Fringe. *J Biol Chem* **282**, 35153-35162, doi:10.1074/jbc.M707040200 (2007).
- 6 Pandey, A. *et al.* Glycosylation of Specific Notch EGF Repeats by O-Fut1 and Fringe Regulates Notch Signaling in Drosophila. *Cell Rep* **29**, 2054-2066 e2056, doi:10.1016/j.celrep.2019.10.027 (2019).
- 7 Takeuchi, H. & Haltiwanger, R. S. Role of glycosylation of Notch in development. *Semin Cell Dev Biol* **21**, 638-645, doi:10.1016/j.semcdb.2010.03.003 (2010).
- 8 Taylor, P. *et al.* Fringe-mediated extension of O-linked fucose in the ligand-binding region of Notch1 increases binding to mammalian Notch ligands. *Proc Natl Acad Sci U S A* **111**, 7290-7295, doi:10.1073/pnas.1319683111 (2014).

Reviewers' Comments:

Reviewer #1:

Remarks to the Author:

I am satisfied with the authors responses and am in favor of publishing the manuscript.

Reviewer #2:

Remarks to the Author:

In this revised draft the authors have made major revisions that fully address my previous concerns. The code is better annotated and more accessible, and they have added convincing new data to the figures. They have also revised the text in accordance with my comments. Thus, I now fully support publication.

Reviewer #3:

Remarks to the Author:

In the revised manuscript, the authors performed additional experiments to address the reviewers' concerns. Although the manuscript has been significantly improved, there are still a few minor issues that need to be addressed.

1. In the revised Figure 6e, the authors nicely summarize the biosynthetic pathways for GlcNAc modification. I would like to point out that it is a common practice in the field of glycobiology to use the blue rectangle for GlcNAc, the green circle for mannose and the blue circle for glucose, and the red triangle for fucose. The current way of showing glycan structures is rather confusing for readers.
2. Related to the comment above, since EOGT is included in the revised Figure 6e, related information and references should be provided in the main text (e.g., Sakaidani et al. Nature Communications 2011).
3. In the revised extended data Fig. 11, the position of the molecular weight markers is missing and should be provided.
4. Related to the comment above, I found it surprising that the authors used HRP-conjugated anti-mouse IgG antibody (Abcam #ab205719) to detect CTD110.6 O-GlcNAc antibody, since the O-GlcNAc antibody is IgM, not IgG.

REVIEWERS' COMMENTS

Reviewer #1 (Remarks to the Author):

I am satisfied with the authors responses and am in favor of publishing the manuscript.

Reviewer #2 (Remarks to the Author):

In this revised draft the authors have made major revisions that fully address my previous concerns. The code is better annotated and more accessible, and they have added convincing new data to the figures. They have also revised the text in accordance with my comments. Thus, I now fully support publication.

Reviewer #3 (Remarks to the Author):

In the revised manuscript, the authors performed additional experiments to address the reviewers' concerns. Although the manuscript has been significantly improved, there are still a few minor issues that need to be addressed.

1. In the revised Figure 6e, the authors nicely summarize the biosynthetic pathways for GlcNAc modification. I would like to point out that it is a common practice in the field of glycobiology to use the blue rectangle for GlcNAc, the green circle for mannose and the blue circle for glucose, and the red triangle for fucose. The current way of showing glycan structures is rather confusing for readers.

Authors' response: we thank you for kind suggestions and modified accordingly.

2. Related to the comment above, since EOGT is included in the revised Figure 6e, related information and references should be provided in the main text (e.g., Sakaidani et al. Nature Communications 2011).

Authors' response: we added suggested information and reference.

3. In the revised extended data Fig. 11, the position of the molecular weight markers is missing and should be provided.

Authors' response: we added the molecular weight markers according to your kind suggestion in the revised Supplementary Fig. 11 (original extended data Fig. 11).

4. Related to the comment above, I found it surprising that the authors used HRP-conjugated anti-mouse IgG antibody (Abcam #ab205719) to detect CTD110.6 O-GlcNAc antibody, since the O-GlcNAc antibody is IgM, not IgG.

Authors' response: we used secondary antibody from Abclonal. We apologize for this error in the

method and thank you for kindly pointing it out.